# SaTeen: Learning Structural Alignment for Continual Test-Time Adaptation

**Chang Liu** [1 2]  **Ruotong Zhao** [1 2]  **Li Gao** [1 2]  **Yupei Zhang** [1 2]

## Abstract

Test-Time Adaptation (TTA) aims to reconcile model generalization in the presence of distribution shifts. Current TTA methods usually leverage sample uncertainty to select reliable samples for model adjustment via entropy minimization (EM). However, sample uncertainty often relies on a plausible metric and leaves many unreliable samples in the EM process, potentially leading to model collapse. Importantly, these excluded samples incur biased data features of the shifted distribution in TTA. This paper introduces SaTeen, a **S**tructural **A**lignment-based **Te**st-Tim**e** Adaptatio**n** method that performs two-fold alignment the structures of test samples with the reliable reference structures. Specifically, the two-fold alignments are 1) Intra-sample structure alignment, where SaTeen maximizes cross-entropy discrepancy between a sample (reference) and its structure-disrupted counterpart, with the assumption of stable dominant features; 2) Inter-sample structure alignment, where SaTeen minimizes the reconstruction error of test samples in the reference subspace spanned by the Incremental PCA on reliable samples, with the assumption of stable intrinsic data manifold. Our extensive experiments demonstrate that SaTeen achieves the state-of-the-art performance across various scenarios for both TTA and continual TTA.

## 1. Introduction

Deep neural networks have achieved great success across various tasks (LeCun et al., 2015; He et al., 2016). However, such success relies on the assumption that data are independent and identically distributed (i.i.d.), which is often violated in real-world scenarios where inputs follow

[1]School of Computer Science, Northwestern Polytechnical University, Xi'an, Shaanxi, China [2]MIIT Lab of Big Data Storage and Management, Xi'an, Shaanxi, China. Correspondence to: Yupei Zhang <ypzhaang@nwpu.edu.cn>.

*Proceedings of the $43^{rd}$ International Conference on Machine Learning*, Seoul, South Korea. PMLR 306, 2026. Copyright 2026 by the author(s).

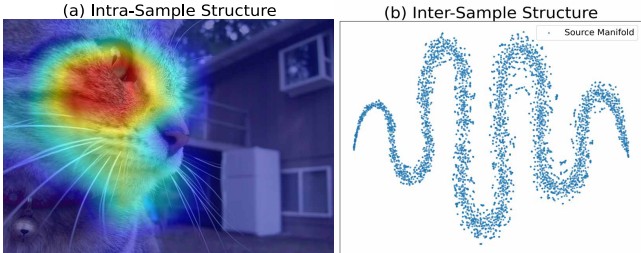

*Figure 1.* (a) Illustration of intra-sample structure, such as shapes, which is usually held in TTA. (b) Illustration of inter-sample structure, i.e. intrinsic manifold, to which test data is generally close. Considering both priors can boost model adaptation smoothly.

distributions different from the training data, leading to compromised performance (Yang et al., 2022; Wu et al., 2024; Wang et al., 2025; Koh et al., 2021; Hu et al., 2024). Test-time adaptation (TTA) (Sun et al., 2020; Ma et al., 2025a; Hu et al., 2025) has emerged as a practical paradigm that enables models trained on a source domain to adapt online to unlabeled test data, addressing non-stationary environmental changes by leveraging each test sample only once for immediate adaptation (Bartler et al., 2022).

A primary cause of performance degradation under distribution shift is that the pre-trained model may over-attend to noisy features while down-weighted core structural features (Zhou et al., 2021; Shi et al., 2022). Observing that model precision decreases on high-entropy samples, Tent (Wang et al., 2021) uses predictive entropy as a proxy for distribution shift and applies entropy minimization (EM) (Zhang et al., 2025a) to reduce noisy-feature effects. However, many test samples still rely heavily on noisy features, potentially degrading model adaptation (Malinin & Gales, 2018; Chan et al., 2020). Subsequently, SAR (Niu et al., 2023) filters out samples that induce large gradients, and DeYO (Lee et al., 2024) removes samples whose predictions rely heavily on noisy features. However, EM-based adaptation methods are prone to model collapse (Han et al., 2025; Chen et al., 2026). Under distribution shifts, the pre-trained model may exhibit imbalanced predictions (Su et al., 2023; Tejero-de Pablos et al., 2024), i.e., assigning a large fraction of samples to a single class. Since the gradient of entropy minimization tends to increase the logit of the most probable class (Agarwal et al., 2026), such gradients accumulate under imbalanced predictions, driving the model to

predict the same class for most inputs and eventually leading to model collapse (Ma et al., 2025b).

To mitigate model collapse and further encourage attention to structural features, we propose an **intra-sample structural alignment** objective by maximizing the predictive cross-entropy between an input and its structure-disrupted counterpart, as shown in Fig. 1. Our theoretical analysis and empirical results demonstrate that this objective 1) promotes learning structural features while suppressing noisy features, and 2) structurally prevents the trivial solution of collapsing all predictions to a single class.

On the other hand, intra-sample alignment TTA is essentially a form of local posterior update that relies solely on the current test data. This local reliance can 1) lead to overfitting to local samples and make the model prone to forgetting previously seen samples (Liang et al., 2025), and 2) encourage learning domain-specific rather than domain-shared features, making the model fragile in continual distribution-shift scenarios (Arjovsky et al., 2019; Zhao et al., 2019; Geirhos et al., 2020). Hence, an effective TTA method should go beyond local prediction-level alignment and explicitly preserve the structural consistency across domains. For example, ViDA (Liu et al., 2024b) preserves consistency by explicitly decomposing representation learning into a low-rank, domain-shared subspace and a high-rank, domain-specific subspace, while a homeostatic gating mechanism prevents instance-wise updates from distorting long-term structure but introduces additional trainable parameters at test time, which slows inference and reduces deployment efficiency. LinearTCA (You et al., 2025) preserves consistency by aligning statistics of test embeddings to source domain via a global linear transformation. However, LinearTCA assumes access to global test statistics, which is incompatible with realistic TTA scenarios where samples arrive sequentially.

To mitigate catastrophic forgetting and local overfitting without introducing additional trainable parameters or disrupting the TTA setup, we propose to perform **inter-sample structural alignment** using Incremental PCA (IPCA) (Ross et al., 2008), as shown in Fig. 1. Specifically, we employ reliable samples to build a PCA (Maćkiewicz & Ratajczak, 1993) subspace, and align all test samples toward this subspace by reducing their reconstruction errors. Our theoretical analysis and empirical results demonstrate that this objective can preserve the structural consistency across evolving target domains.

## 2. Preliminaries

### 2.1. Continual Test-Time Adaptation

Considering the classification question, let $\mathcal{X}$ represent the sample space and $\mathcal{Y} = \{1, 2, \ldots, \mathcal{C}\}$ be the set of labels,

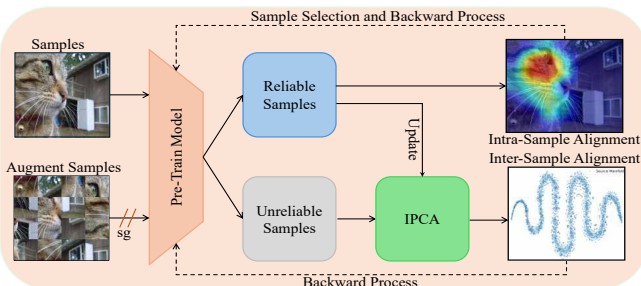

*Figure 2.* Framework of SaTeen. SaTeen uses the intra-sample structural loss ($\mathcal{L}_1(\lambda_1, \mathbf{x})$ in Eq. (6)) to classify samples into reliable and unreliable samples. Reliable samples participate in both intra-sample and inter-sample alignment, while unreliable samples only participate in Inter-sample alignment.

where $\mathcal{C}$ is the number of classes. The training samples are assumed to be independently and identically distributed (i.i.d.) from a joint distribution $P_S(\mathbf{x}, y)$ over $\mathcal{X} \times \mathcal{Y}$, a.k.a. the **S**ource-domain distribution. Besides, denote by $P_S(x)$ the marginal distribution over $\mathcal{X}$. Under covariate shift in TTA, the marginal distribution over the input space changes from source to the **T**arget domain, while the conditional distribution of each label $y$ remains invariant, i.e., $P_S(\mathbf{x}) \neq P_T(\mathbf{x})$ while $P_S(y \mid \mathbf{x}) = P_T(y \mid \mathbf{x})$.

TTA aims to adapt the model $P_\theta(y|\mathbf{x})$ pretrained on the source-domain distribution $P_S(\mathbf{x}, y)$ to multiple test target-domain distributions $\{\mathcal{P}_{T_i}(\mathbf{x})\}_{i=1}^n$, where $n$ indicates the scale of target-domain distributions. Continual TTA (Wang et al., 2022) follows an online learning paradigm: At time step $t$, only a mini-batch of unlabeled samples $\mathbf{B}_t = \{\mathbf{x}_{ti}\}_{i=1}^{N_{bs}}$ is available, where $N_{bs}$ denotes the scale of a mini-batch and $\mathbf{x}_{ti}$ is drawn from $\{\mathcal{P}_{T_i}(\mathbf{x})\}_{i=1}^n$.

**Notice**: The intractable conditions of continual TTA are two-fold. The model cannot access any source-domain samples and can only exploit unlabeled target-domain samples once. And the target-domain distributions continue to change over time.

### 2.2. Entropy Minimization & Model Collapse

Shannon entropy (Shannon, 1948) is a common metric that measures the uncertainty of an information source, where higher entropy indicates greater uncertainty in its underlying probability distribution. For a specific sample $\mathbf{x} \in \mathcal{X}$, the Shannon entropy is defined as,

$$H(\mathbf{p}(\mathbf{x})) = -\sum_{i=1}^{\mathcal{C}} p_i(\mathbf{x}) \log p_i(\mathbf{x}), \qquad (1)$$

where $\mathbf{p}(\mathbf{x})$ represents the probability distribution of the model's output, $p_i(\mathbf{x})$ is the probability that the model assigns the sample to class $i$, *i.e.*, $p_i(\mathbf{x}) = p_\theta(y = i \mid \mathbf{x})$. In

practice, $H(\mathbf{p}(\mathbf{x}))$ can be decoupled into (Ma et al., 2025b):

$$-\sum_{i=1}^{\mathcal{C}} p_i(\mathbf{x})\,u_i \;+\; \log \sum_{i=1}^{\mathcal{C}} e^{u_i}, \qquad (2)$$

where $u_i$ indicates the logit corresponding to class $i$. When it serves as the objective function of model adaptation, the first term of Eq. (2) acts as a reward that reinforces the logit of the currently most probable class, while the second term serves as a penalty that globally regularizes and suppresses the magnitudes of all class logits.

**Definition 2.1. (Imbalanced Prediction Rate)** Let $\delta$ denote the *imbalanced prediction rate* of a pre-trained model when applied to a different target domain. The imbalance prediction rate is defined as

$$\delta = \max_{c \in \mathcal{Y}} \Pr_{\mathbf{x} \sim P_T}\left(\hat{y} = c \mid \mathbf{x}\right), \qquad (3)$$

where $\hat{y}$ is the predicted label and $\delta \in [\frac{1}{\mathcal{C}}, 1]$.

Intuitively, when the $\delta$ is sufficiently large, for most samples, the predicted vectors $\mathbf{p}(\mathbf{x}) = \{p_i(\mathbf{x})\}_{i=1}^{\mathcal{C}}$ concentrate in a single direction, which disrupts the balance between the reward and penalty in Eq. (2). In this case, the gradients of EM induced by these samples can drive the model toward collapse. Empirically, as shown in Fig. 3(a), when distribution shift occurs, the pre-trained model exhibits a high imbalanced prediction rate. Meanwhile, the pseudo intra-class variance of that class is significantly smaller than that in the source domain, shown in Fig. 3(b), indicating that these misclassified samples rely on highly similar features.

**Theorem 2.2. (Imbalanced Prediction Induces Model Collapse)** *With specific assumptions, if the $\delta$ is sufficiently large, then there exists a parameter-update direction $\mathbf{v} \in \mathbb{R}^{|\Phi|}$, where $\Phi$ denotes the model parameter space. The following properties hold:*

*(1) For a sample $\mathbf{x}$, its likelihood on $c$, i.e, $p_c(\mathbf{x})$, increases monotonically along $\mathbf{v}$;*

*(2) For a sample $\mathbf{x}$, when $p_c$ becomes sufficiently large, the Hessian of the entropy $H(\mathbf{p}(\mathbf{x}))$ attains its minimum eigenvalue along the collapse direction $\mathbf{v}$, namely*

$$\lambda_{\min}\!\left(\nabla^2 H(\mathbf{p}(\mathbf{x}))\right) = \mathbf{v}^\top \nabla^2 H(\mathbf{p}(\mathbf{x}))\,\mathbf{v} \le 0. \quad (4)$$

*(3) On the target-domain distribution, the EM gradient and the parameter update have a positive projection onto $\mathbf{v}$, i.e.,*

$$\mathbb{E}_{\mathbf{x} \sim \mathcal{P}_T(\mathbf{x})}\left[\langle \nabla H(\mathbf{p}(\mathbf{x})),\, \mathbf{v}\rangle\right] > 0. \qquad (5)$$

*Proof*: The proofs can be found in Appendix A.

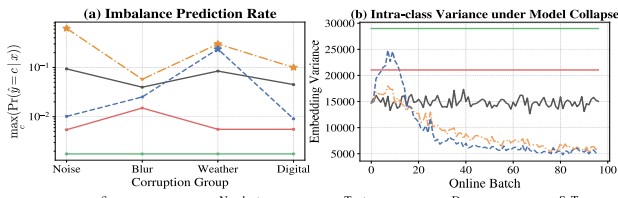

*Figure 3.* (a) The imbalance prediction rate $\delta$ under the single TTA scenario on ImageNet-C for different methods, as defined in Eq. (3). When $\delta$ approaches 1, it indicates model collapse, which is marked with **\***. (b) The pseudo intra-class variance change of the imbalanced predicted class $c$ under the test flow during model collapse on 'fog' corruption. It is observed that, as model collapse occurs, a large number of samples from other classes are predicted as class $c$ and the pseudo intra-class variance rapidly decreases, falling significantly below the pre-trained model's intra-class variance for that class.

**Takeaway**: Let us consider the type of noisy features that pollute most test samples. When the model is trained to overfit the noisy features, implying error accumulation, there exists a parameter direction $\mathbf{v}$ that is coupled with the weights of noisy features. From Theorem 2.2, increasing these weights monotonically increases the predicted probability $p_c$ as well $\delta$, leading to model collapse.

## 3. Method

### 3.1. Intra-Sample Structure Alignment

To address the high prediction imbalance in EM-based TTA methods caused by its over-reliance on noisy features, this study encourages the model to focus on the kernel structural features of the samples. Inspired by contrastive learning (CL) (Chen et al., 2020; Chen & He, 2021), we leverage a negative sample that disrupts structural features of the given sample to reduce similarity between the two versions.

To highlight the intra-sample structure, we adopt the strategy of patch shuffling (Geirhos et al., 2018), which segments the given sample $\mathbf{x}$ into many tokens and then randomly re-assigns them. For image classification, we divide an image into patches and randomly shuffling them to disrupt object's shapes. Then, using the structure-disrupted counterpart sample $\tilde{\mathbf{x}}$ as the negative sample, our CL objective is formulated as:

$$\mathcal{L}_1(\lambda_1, \mathbf{x}) = H(\mathbf{p}(\mathbf{x})) - \lambda_1\,\mathrm{CE}(sg[\mathbf{p}(\tilde{\mathbf{x}})] \,\|\, \mathbf{p}(\mathbf{x})), \quad (6)$$

where $\tilde{\mathbf{x}}$ denotes the negative sample, $\mathrm{CE}(\cdot)$ represents the cross-entropy loss, $\lambda_1$ is the balance parameter of the cross-entropy term, and sg$[\cdot]$ denotes the stop-gradient operator. Similarly, we can decouple the $\mathcal{L}_1(\lambda_1, \mathbf{x})$loss into the following form:

$$-\sum_{i=1}^{\mathcal{C}} \left(p_i(\mathbf{x}) - \lambda_1\,p_i(\tilde{\mathbf{x}})\right) u_i + (1-\lambda_1)\log\sum_{i=1}^{\mathcal{C}} e^{u_i}. \quad (7)$$

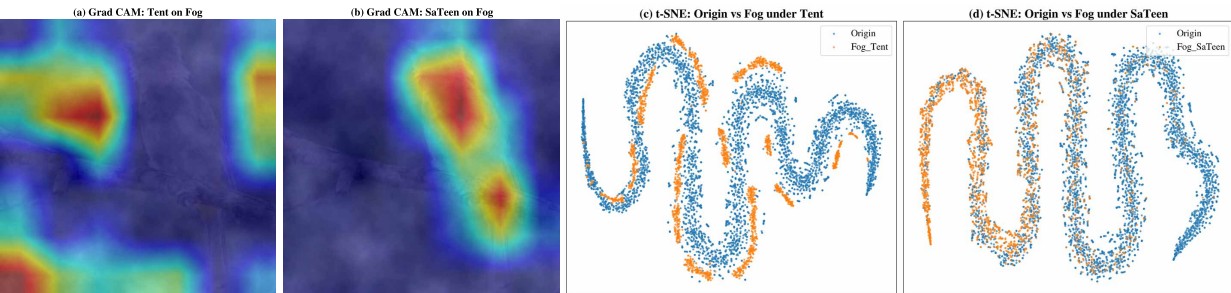

**Figure 4.** Visualizations of structure alignment. (a) and (b) show the Grad-CAM visualizations of a 'fog' image from ImageNet-C after Tent (Wang et al., 2021) and SaTeen adaptation, respectively. The results indicate that under severe distribution shift, after Tent adaptation, the model incorrectly attends to the background rather than the structural features of the object, whereas after SaTeen adaptation, the model focuses more on the structural features of the object. (c) and (d) present the t-SNE visualizations of ImageNet-val and ImageNet-C after Tent (Wang et al., 2021) and SaTeen adaptation, respectively. The results show that after Tent adaptation, samples from the target and source domains can be clearly separated in low-dimensional space, while after SaTeen adaptation, samples from the two domains become difficult to distinguish, indicating that SaTeen preserves structural consistency across domains.

The detailed derivation can be found in the Appendix A. Intuitively, this loss introduces a dynamic suppression effect on the reward term in Eq. (2). When the predictions of a sample and the negative are similar, the reward is effectively attenuated.

**Theorem 3.1.** *(The $\mathcal{L}_1(\lambda_1, \mathbf{x})$ Loss Mitigates Model Collapse) For the target-domain distribution that the imbalance prediction rate is $\delta$, if $\lambda_1$ is sufficiently large, then the $\mathcal{L}_1(\lambda_1, \mathbf{x})$-loss gradient and the parameter update has a non-positive projection onto $\mathbf{v}$, i.e.,*

$$\mathbb{E}_{\mathbf{x} \sim \mathcal{P}_T(\mathbf{x})}\left[\langle \nabla \mathcal{L}_1(\lambda_1, (\mathbf{x})), \mathbf{v} \rangle\right] \leq 0. \tag{8}$$

*Proof*: The proofs can be found in Appendix A.

**Takeaway**: Theorem 3.1 implies that, when $\lambda_1$ is sufficiently large, the $\mathcal{L}_1(\lambda_1, \mathbf{x})$ loss prevents the accumulation of updates along the direction $\mathbf{v}$ even under imbalanced prediction, thereby avoiding model collapse. A detailed proof, as well as an explanation of why we do not adopt $CE(\mathbf{p}(\mathbf{x}) \| sg[\mathbf{p}(\tilde{\mathbf{x}})])$, is provided in the Appendix A.

### 3.2. Inter-Sample Structure Alignment

Intra-sample structural alignment encourages the model to pay attention to the kernel features within a sample, while the entire distribution of test samples is more important for model adaptation. This study assumes that the intrinsic subspace of embedded samples is generally maintained between two continuous time steps in TTA.

Hence, we employ the Incremental Principal Component Analysis (IPCA) (Ross et al., 2008) to identify the low-dimensional PCA subspace from the reliable samples of low gradients, where both the model and test samples inhabit and which is warmly shifted along the data stream in continual TTA. Note that IPCA can update this subspace in an online-learning manner to model such warm data shifts.

Specifically, we first accumulate a set of reliable target samples' features, denoted as $\mathbf{B}_0^r = \{\mathbf{z}_i\}_{i=1}^{N_0}$ to capture the IPCA subspace, the set of reliable samples is defined as follows:

$$\Gamma = \{\mathbf{x} \mid \mathcal{L}_1(\lambda_1, \mathbf{x}) < \tau\}, \tag{9}$$

where $\tau$ is a threshold for data belief. The empirical mean is computed as $\boldsymbol{\mu}_0 = \bar{\mathbf{B}}_0^r = \frac{1}{N_0} \sum_{i=1}^{N_0} \mathbf{z}_i$, where $\mu_0$ represents the initial mean estimated from $\hat{\mathbf{B}}_0^r$. Given the mean, we center the initial samples as $\hat{\mathbf{B}}_0^r = \mathbf{B}_0^r - \boldsymbol{\mu}_0$. The initial orthogonal basis is then obtained by performing the singular value decomposition (SVD) on $\hat{\mathbf{B}}_0^r$:

$$\hat{\mathbf{B}}_0^r = \mathbf{U}_0 \boldsymbol{\Sigma}_0 \mathbf{V}_0, \tag{10}$$

where $\mathbf{U}_0 \in R^{d \times d}$, $\boldsymbol{\Sigma}_0 \in R^{d \times N_0}$, and $\mathbf{V}_0 \in R^{N_0 \times N_0}$ denote the left singular value matrix, the diagonal matrix of singular values, and the right singular value matrix, respectively, $d$ is the dimension of the penultimate layer.

As reliable test samples $\mathbf{B}_t^r = \{\mathbf{z}_{ti}\}_{i=1}^{N_t}$ at time step $t-1$, we update the IPCA statistics in an online manner. Specifically, the running mean is updated as follows:

$$\boldsymbol{\mu}_t = \frac{N_{t-1} \boldsymbol{\mu}_{t-1} + N_t \bar{\mathbf{B}}_t^r}{N_{t-1} + N_t}, \tag{11}$$

where $\boldsymbol{\mu}_{t-1}$ is the overall mean feature at time step $t - 1$, and $\bar{\mathbf{B}}_t^r$ is the incoming mean feature at time step $t$. The total number of reliable samples at time $t$ is $N_{t-1} + N_t$.

Considering that the mean estimation error is large under small batch size, we centralize the incoming samples with the overall mean $\hat{\mathbf{B}}_t^r = \mathbf{B}_t^r - \boldsymbol{\mu}_t$. We project $\hat{\mathbf{B}}_t^r$ onto the current subspace by:

$$\mathbf{P}_t = \mathbf{U}_{t-1}^\top \hat{\mathbf{B}}_t^r, \quad \mathbf{Q}_t = \hat{\mathbf{B}}_t^r - \mathbf{U}_{t-1} \mathbf{U}_{t-1}^\top \hat{\mathbf{B}}_t^r. \tag{12}$$

For online learning, the subspace is updated by constructing

a matrix from the projected components:

$$\mathbf{M}_t = \begin{pmatrix} \boldsymbol{\Sigma}_{t-1} & \mathbf{P}_t \\ \mathbf{0} & \text{orth}(\mathbf{Q}_t)\mathbf{Q_t} \end{pmatrix}, \qquad (13)$$

where $\mathbf{M}_t \in R^{(k+N_t) \times (k+N_t)}$, the $k$ is the number of singular values in $\boldsymbol{\Sigma}_{t-1}$, and $\text{orth}(\cdot)$ denotes the orthogonalization operation.

Then we perform the SVD on $\mathbf{M}_t := \tilde{\mathbf{U}}_t \tilde{\boldsymbol{\Sigma}}_t \tilde{\mathbf{V}}_t$, and update the orthogonal basis by:

$$\mathbf{U}_t = [\,\mathbf{U}_{t-1}, \ \text{orth}(\mathbf{Q}_t)\,]\tilde{\mathbf{U}}_t. \qquad (14)$$

Finally, all incoming samples can be aligned to the reliable PCA-feature subspace where the model potentially lies in, by minimizing the following reconstruction error,

$$\mathcal{L}_2(\mathbf{x}) = \left\| \mathbf{z}_{t+1} - \mathbf{U}_t \mathbf{U}_t^\top \mathbf{z}_{t+1} \right\|_2 . \qquad (15)$$

**Theorem 3.2** (Upper-Bounded Source-Domain Orthogonal Basis Estimated from Reliable Target Samples). *Let* $\Omega := \bigcup_{x \sim \mathcal{P}_T} \mathcal{B}(x, r^*)$ *be the set of balls near the test data, where* $\mathcal{B}(x, r^*) = \{\mathbf{x}_0 : \|\mathbf{x}_0 - \mathbf{x}\| \leq r^*\}$. *We sample* $N_t$ *source samples from* $\mathcal{P}_S \cap \Omega$ *and* $N_t$ *target samples from* $\mathcal{P}_T$. *Let* $\mathbf{U}_S$ *and* $\mathbf{U}_T$ *denote the orthogonal bases. When Assumptions A.1, A.2, and A.3 hold, with a probability of* $1 - \epsilon_p$ *we have :*

$$\|\sin \Theta(\mathbf{U}_S, \mathbf{U}_T)\|_F \leq \epsilon_{\mathbf{u}}, \qquad (16)$$

*where* $\epsilon_{\mathbf{u}}$ *denotes an upper bound on the source-domain covariance estimated from reliable target samples, which is related to the prediction confidence and the prediction error of the sampled samples.*

Based on Theorems 3.2, we can reduce the estimation error of orthogonal basis by selecting reliable samples with low entropy and a high likelihood of being correctly predicted. IPCA (Ross et al., 2008) proves that the orthogonal basis of the target-domain covariance matrix can be computed via the incremental procedure described above. This enables aligning newly arriving target-domain samples to the source domain in an online manner, which helps preserve structural consistency as the target domain evolves.

### 3.3. The Proposed SaTeen

Based on the above intra- and inter-sample structure alignments, we propose a **S**tructural **A**lignment-based **Te**st-Time Adaptatio**n** method, dubbed SaTeen for short. The framework of SaTeen can be found in Fig. 2.

SaTeen performs sample selection and weighting based on $\mathcal{L}_1(\lambda_1, \mathbf{x})$ in Eq. (6) . Considering that $H(\mathbf{p}(\mathbf{x})) \in [0, \log \mathcal{C}]$ while $\text{CE}(sg[\mathbf{p}(\tilde{\mathbf{x}})] \,\|\, \mathbf{p}(\mathbf{x})) \in [0, +\infty)$, to avoid

*Table 1.* Comparisons with baselines on ColoredMNIST and WaterBirds regarding accuracy (%).

| ColoredMNIST | Avg | Worst-Group |
|---|---|---|
| ResNet18-BN | $65.3_{\pm 0.1}$ | $20.1_{\pm 0.1}$ |
| Tent (Wang et al., 2021) | $59.8_{\pm 0.2}$ | $13.0_{\pm 0.7}$ |
| SENTRY (Prabhu et al., 2021) | $63.2_{\pm 0.4}$ | $13.0_{\pm 0.8}$ |
| EATA (Niu et al., 2022) | $63.4_{\pm 0.2}$ | $20.8_{\pm 0.2}$ |
| SAR (Niu et al., 2023) | $61.3_{\pm 0.2}$ | $15.6_{\pm 0.5}$ |
| DeYO (Lee et al., 2024) | $\underline{76.4}_{\pm 0.3}$ | $\underline{52.7}_{\pm 0.4}$ |
| **SaTeen (Ours)** | $\mathbf{79.3}_{\pm 0.3}$ | $\mathbf{67.2}_{\pm 0.3}$ |

| Waterbirds | Avg | Worst-Group |
|---|---|---|
| ResNet50-BN | $84.3_{\pm 0.1}$ | $67.0_{\pm 0.1}$ |
| Tent (Wang et al., 2021) | $83.2_{\pm 0.6}$ | $54.5_{\pm 2.0}$ |
| SENTRY (Prabhu et al., 2021) | $85.7_{\pm 0.6}$ | $61.0_{\pm 1.4}$ |
| EATA (Niu et al., 2022) | $83.0_{\pm 0.4}$ | $54.0_{\pm 1.3}$ |
| SAR (Niu et al., 2023) | $83.1_{\pm 0.5}$ | $53.8_{\pm 1.4}$ |
| DeYO (Lee et al., 2024) | $\underline{88.2}_{\pm 0.7}$ | $\underline{74.6}_{\pm 1.3}$ |
| **SaTeen (Ours)** | $\mathbf{89.0}_{\pm 0.6}$ | $\mathbf{76.3}_{\pm 1.5}$ |

*Table 2.* Comparisons with baselines on ImageNet-C at severity levels under a mixture of 15 corruptions over accuracy (%).

| Mixed Shifts | Lev. 5 | Lev. 3 | Avg. |
|---|---|---|---|
| ResNet50-GN | $30.6_{\pm 0.1}$ | $54.0_{\pm 0.1}$ | $42.3_{\pm 0.1}$ |
| Tent (Wang et al., 2021) | $34.2_{\pm 1.0}$ | $33.1_{\pm 0.1}$ | $33.7_{\pm 0.6}$ |
| EATA (Niu et al., 2022) | $38.2_{\pm 0.4}$ | $56.1_{\pm 0.1}$ | $47.2_{\pm 0.3}$ |
| SAR (Niu et al., 2023) | $38.3_{\pm 0.5}$ | $57.4_{\pm 0.1}$ | $47.9_{\pm 0.3}$ |
| DeYO (Lee et al., 2024) | $38.6_{\pm 1.3}$ | $\underline{59.2}_{\pm 0.1}$ | $48.9_{\pm 0.7}$ |
| ReCAP (Hu et al., 2025) | $\underline{41.7}_{\pm 0.7}$ | $59.0_{\pm 0.1}$ | $\underline{50.4}_{\pm 0.4}$ |
| **SaTeen (Ours)** | $\mathbf{43.7}_{\pm 0.6}$ | $\mathbf{60.5}_{\pm 0.1}$ | $\mathbf{52.1}_{\pm 0.4}$ |
| VitBase-LN | $29.9_{\pm 0.1}$ | $53.8_{\pm 0.1}$ | $41.9_{\pm 0.1}$ |
| Tent (Wang et al., 2021) | $24.1_{\pm 0.1}$ | $70.2_{\pm 0.1}$ | $47.2_{\pm 0.1}$ |
| EATA (Niu et al., 2022) | $56.4_{\pm 0.1}$ | $69.6_{\pm 0.1}$ | $63.0_{\pm 0.1}$ |
| SAR (Niu et al., 2023) | $57.1_{\pm 0.1}$ | $70.7_{\pm 0.1}$ | $63.9_{\pm 0.1}$ |
| DeYO (Lee et al., 2024) | $59.4_{\pm 0.1}$ | $\underline{72.1}_{\pm 0.1}$ | $\underline{65.8}_{\pm 0.1}$ |
| ReCAP (Hu et al., 2025) | $59.4_{\pm 0.1}$ | $71.8_{\pm 0.1}$ | $65.6_{\pm 0.1}$ |
| **SaTeen (Ours)** | $\mathbf{60.3}_{\pm 0.1}$ | $\mathbf{72.3}_{\pm 0.1}$ | $\mathbf{66.3}_{\pm 0.1}$ |

excessively large weights for certain samples, we assign each sample a weight

$$\alpha(\mathbf{x}) = \exp(\min\left(-\mathcal{L}_1(\lambda_1, \mathbf{x}), \lambda_1 \log \mathcal{C}\right)). \qquad (17)$$

Finally, our overall sample-weighted loss is given by

$$\mathcal{L}_{\text{SaTeen}}(\mathbf{x}) = \alpha(\mathbf{x}) \left( \mathbb{I}_{(\mathbf{x} \in \Gamma)} \mathcal{L}_1(\lambda_1, \mathbf{x}) + \lambda_2 \mathcal{L}_2(\mathbf{x}) \right), \quad (18)$$

where $\mathbb{I}(\cdot)$ is the indicator function, $\Gamma$ is defined in Eq. (9), and $\lambda_2$ is a balance parameter.

## 4. Experiment

### 4.1. Experimental Settings

**Datasets and Methods** For test datasets, we employ: 1) ImageNet-C (Hendrycks & Dietterich, 2019), a widely used TTA benchmark comprising 15 corruption types, each with 5 severity levels; and 2) ColoredMNIST and WaterBirds,

*Table 3.* Accuracy (%) under single TTA on ImageNet-C across corruption types at severity level 5.

| Single TTA | Noise | | | Blur | | | | Weather | | | | Digital | | | | Avg. |
|---|---|---|---|---|---|---|---|---|---|---|---|---|---|---|---|---|
| | Gauss. | Shot | Impl. | Defoc. | Glass | Motion | Zoom | Snow | Frost | Fog | Brit. | Contr. | Elastic | Pixel | JPEG | |
| ResNet50-GN | 15 | 19.8 | 17.9 | 19.7 | 11.3 | 21.4 | 24.9 | 40.4 | 47.3 | 33.6 | 69.3 | 36.3 | 18.6 | 28.4 | 52.2 | 30.6 |
| Tent (Wang et al., 2021) | 5.4 | 6.1 | 5.8 | 14.8 | 6.5 | 21.8 | 22.2 | 25.8 | 33.6 | 3.7 | 70.5 | 42.4 | 11.7 | 48.2 | 54.3 | 33.5 |
| EATA (Niu et al., 2022) | 28.9 | 30.9 | 31.2 | 19.7 | 19.0 | 31.0 | 32.1 | 41.6 | 42.4 | 20.3 | 70.1 | 43.7 | 16.9 | 49.7 | 56.2 | 36.9 |
| SAR (Niu et al., 2023) | 28.4 | 31.6 | 29.9 | 18.5 | 18.1 | 30.8 | 30.4 | 42.0 | 43.6 | 5.2 | 70.7 | 44.0 | 17.0 | 48.7 | 55.2 | 34.3 |
| DeYO (Lee et al., 2024) | 39.0 | 41.9 | 40.9 | 21.9 | 23.7 | 38.7 | _37.6_ | 50.8 | 49.3 | 3.5 | 73.1 | 50.1 | 42.0 | 56.1 | 57.7 | 42.4 |
| ReCAP (Hu et al., 2025) | 40.5 | 42.7 | 41.8 | 21.4 | 2.7 | 40.1 | **41.1** | 16.8 | 48.6 | 56.2 | 72.9 | 50.5 | 3.5 | 56.3 | 57.8 | 39.5 |
| **SaTeen (Intra-only)** | _41.1_ | _44.0_ | _42.9_ | _25.3_ | _25.7_ | _41.2_ | 37.5 | _53.5_ | _51.6_ | _57.7_ | _73.2_ | _52.3_ | _46.1_ | _58.6_ | _58.8_ | _47.1_$_{\pm 0.2}$ |
| **SaTeen (Intra and Inter)** | 42.2 | 45.4 | 44.3 | 32.4 | 29.9 | 44.0 | 41.1 | 56.6 | 53.9 | 60.3 | 73.7 | 54.0 | 49.9 | 60.4 | 59.8 | **49.9**$_{\pm 0.2}$ |
| VitBase-LN | 9.4 | 6.7 | 8.3 | 29.1 | 23.4 | 34.0 | 27.1 | 15.8 | 26.4 | 47.4 | 54.7 | 44.0 | 30.5 | 44.5 | 47.6 | 29.9 |
| Tent (Wang et al., 2021) | 42.3 | 1.7 | 43.4 | 51.9 | 47.9 | 55.1 | 50.1 | 19.3 | 20.5 | 66.4 | 74.7 | 64.7 | 52.1 | 66.7 | 64.1 | 48.1 |
| EATA (Niu et al., 2022) | 44.1 | 15.0 | 46.1 | 53.2 | 49.7 | 54.9 | 52.6 | 55.3 | 55.4 | 67.1 | 73.7 | 64.6 | 55.4 | 66.9 | 64.3 | 54.5 |
| SAR (Niu et al., 2023) | 44.3 | 13.0 | 45.5 | 52.9 | 50.0 | 55.7 | 51.4 | 56.4 | 55.2 | 66.4 | 74.7 | 64.4 | 55.1 | 66.7 | 64.0 | 54.4 |
| DeYO (Lee et al., 2024) | 53.0 | 53.2 | 53.8 | 58.1 | 58.5 | 63.2 | 58.5 | 67.0 | 65.8 | _73.4_ | _78.2_ | _68.1_ | 67.7 | 73.2 | 70.0 | 64.1 |
| ReCAP (Hu et al., 2025) | 47.6 | 45.7 | 46.0 | 56.2 | 55.3 | 60.8 | 57.0 | 63.7 | 63.5 | 72.0 | 77.3 | 67.3 | 63.8 | 71.6 | 68.6 | 61.5 |
| **SaTeen (Intra-only)** | _54.3_ | _55.0_ | _55.5_ | _58.8_ | _59.4_ | _64.6_ | _60.3_ | _68.3_ | _66.4_ | _73.6_ | _78.2_ | _68.3_ | _68.9_ | _73.9_ | _70.6_ | _65.1_$_{\pm 0.1}$ |
| **SaTeen (Intra and Inter)** | 54.7 | 55.4 | 55.6 | 59.2 | 59.9 | 64.9 | 61.9 | 68.5 | 66.6 | 73.6 | 78.4 | _68.1_ | 69.1 | 74.1 | 70.7 | **65.4**$_{\pm 0.1}$ |

*Table 4.* Accuracy (%) of continual TTA on ImageNet-C across corruption types at severity level 5.

| Continual TTA | Noise | | | Blur | | | | Weather | | | | Digital | | | | Avg. |
|---|---|---|---|---|---|---|---|---|---|---|---|---|---|---|---|---|
| | Gauss. | Shot | Impl. | Defoc. | Glass | Motion | Zoom | Snow | Frost | Fog | Brit. | Contr. | Elastic | Pixel | JPEG | |
| ResNet50-GN | 18.0 | 19.8 | 17.9 | 19.8 | 11.3 | 21.4 | 24.9 | 40.4 | _47.3_ | 33.6 | 69.3 | 36.3 | 18.6 | 28.4 | 52.3 | 30.6 |
| Tent (Wang et al., 2021) | 16.2 | 5.5 | 0.9 | 12.6 | 8.1 | 10.2 | 7.5 | 15.0 | 4.4 | 1.8 | 16.6 | 2.5 | 0.2 | 1.7 | 8.2 | 8.2 |
| EATA (Niu et al., 2022) | 22.1 | 32.7 | 36.9 | 18.9 | 20.9 | 23.1 | 26.3 | 39.9 | 44.6 | 45.7 | 67.6 | 42.3 | 24.1 | 42.6 | 54.9 | 36.2 |
| SAR (Niu et al., 2023) | 23.1 | 33.5 | 37.3 | 18.6 | 19.9 | 22.9 | 26.0 | 39.4 | 43.6 | 44.7 | 67.1 | 41.5 | 23.7 | 42.0 | 54.0 | 35.8 |
| DeYO (Lee et al., 2024) | 28.0 | _41.1_ | _43.6_ | 17.2 | 21.2 | 24.7 | 31.7 | 42.0 | 47.1 | 51.0 | 70.0 | 48.2 | 27.2 | 51.6 | 56.8 | 40.1 |
| ReCAP (Hu et al., 2025) | 19.1 | 24.7 | 27.3 | 20.4 | 15.9 | 26.5 | 29.6 | 42.8 | 44.3 | 48.5 | 70.0 | 44.3 | 22.0 | 45.0 | 55.9 | 36.4 |
| **SaTeen (Intra-only)** | **37.1** | 39.8 | 42.9 | 20.6 | _22.6_ | 36.8 | 35.4 | _47.3_ | 47.1 | _53.9_ | 72.5 | _48.8_ | _36.6_ | _54.1_ | _56.9_ | _43.2_$_{\pm 1.2}$ |
| **SaTeen (Intra and Inter)** | _35.0_ | 47.8 | 47.9 | 27.1 | 30.1 | 38.8 | 38.8 | 48.9 | 53.5 | 57.9 | 70.1 | 51.7 | 41.1 | 58.3 | 59.4 | **47.1**$_{\pm 1.0}$ |
| VitBase-LN | 47.0 | 48.2 | 47.9 | 31.5 | 21.2 | 41.5 | 36.7 | 50.1 | 45.8 | 42.3 | 73.6 | 8.6 | 42.5 | 62.0 | 63.8 | 44.2 |
| Tent (Wang et al., 2021) | 47.8 | 51.1 | 50.8 | 34.2 | 27.0 | 45.5 | 41.6 | 56.0 | 52.3 | 49.7 | 76.1 | 27.2 | 45.3 | 65.6 | 66.1 | 49.0 |
| CoTTA (Wang et al., 2022) | 47.1 | 48.4 | 48.6 | 31.7 | 21.9 | 42.9 | 38.0 | 51.8 | 47.3 | 44.7 | 74.1 | 10.0 | 43.6 | 63.6 | 64.8 | 45.2 |
| EATA (Niu et al., 2022) | 50.3 | 55.8 | 55.3 | 42.5 | 40.2 | 54.6 | 49.2 | 59.1 | 56.2 | 59.0 | 75.9 | 42.9 | 51.1 | 67.5 | 67.9 | 55.1 |
| SAR (Niu et al., 2023) | 50.7 | 56.2 | 55.1 | 41.8 | 39.1 | 53.9 | 48.2 | 58.7 | 55.9 | 58.2 | 76.2 | 42.8 | 50.1 | 67.1 | 67.3 | 54.8 |
| DeYO (Lee et al., 2024) | 53.0 | _57.7_ | _58.4_ | 54.1 | 58.1 | _61.6_ | 35.5 | 62.4 | 63.2 | 70.4 | 76.8 | 64.7 | 67.2 | _71.7_ | 69.3 | 61.6 |
| ViDA (Liu et al., 2024b) | 52.3 | 57.5 | 57.1 | 47.8 | 43.1 | 54.5 | 51.1 | 61.1 | 57.3 | 59.3 | 75.7 | 47.2 | 50.9 | 66.5 | 66.9 | 56.6 |
| CMAE (Liu et al., 2024a) | _53.7_ | 58.1 | 57.5 | 48.6 | 45.1 | 56.7 | _59.3_ | _65.8_ | 64.2 | 35.7 | 76.6 | 39.7 | 62.5 | 70.8 | 68.6 | 57.5 |
| ReCAP (Hu et al., 2025) | 37.9 | 47.8 | 52.9 | 49.1 | 50.9 | 56.3 | 53.1 | 58.4 | 61.5 | 65.8 | 76.3 | 62.8 | 57.9 | 67.3 | 67.2 | 57.7 |
| REM (Han et al., 2025) | **56.5** | **61.9** | **60.8** | 46.8 | 51.0 | 56.5 | 57.2 | 62.5 | _64.8_ | 64.6 | 76.8 | 53.2 | 58.4 | 71.1 | _69.8_ | 60.8 |
| **SaTeen (Intra-only)** | 51.1 | 56.8 | 58.0 | _54.5_ | _58.1_ | 61.3 | 54.1 | 62.7 | 64.2 | _70.5_ | _76.9_ | _65.4_ | _66.0_ | 71.4 | 69.2 | _62.7_$_{\pm 0.8}$ |
| **SaTeen (Intra and Inter)** | 53.6 | 53.8 | 54.4 | 58.7 | 59.2 | 63.9 | 60.5 | 67.5 | 66.1 | 73.0 | 78.1 | 68.6 | 67.9 | 73.5 | 70.1 | **64.6**$_{\pm 0.5}$ |

two benchmarks with extreme spurious correlation shifts. We conduct single TTA and continual TTA experiments on ImageNet-C under mild and wild (e.g. imbalance labels) scenarios. Besides, we include spurious correlation benchmarks on ColoredMNIST and Waterbirds to evaluate robustness under shortcut features. We select Tent (Wang et al., 2021), SENTRY (Prabhu et al., 2021), EATA (Niu et al., 2022), CoTTA (Wang et al., 2022), SAR (Niu et al., 2023), CMAE (Liu et al., 2024a), DeYO (Lee et al., 2024), ViDA (Liu et al., 2024b), REM (Han et al., 2025), Re-CAP (Hu et al., 2025) as the baseline for comparison.

**Models and Implementation Details** We conduct experiments on ResNet18-BN, ResNet50-GN and VitBase-LN that are widely used in TTA studies. For our SaTeen, we use SGD with a momentum of 0.9, and a batch size of 64 (except for the experiments of batch size = 1), and learning rate of 0.00025/0.001 for ResNet/ViT models under single TTA scenario and 0.001/0.001 for continual TTA scenario. $\lambda_1$ and $\lambda_2$ in Eq. (18) are set to 0.4 and 0.05, respectively; the threshold $\tau$ is set to $0.2 \times logC$; $k$ is set to 64. For

trainable parameters of SaTeen during TTA, we adapt the affine parameters of batch/group/layer normalization layers in ResNet18-BN/ResNet50-GN/VitBase-LN, similar to the Tent (Wang et al., 2021). More details and parameter settings can be found in Appendix D. The code is available at https://github.com/scienceliuc/SaTeen.

### 4.2. Main Results

**Biased Scenario** We present the evaluation of SaTeen under a biased scenario characterized by spurious correlations, conducted on the ColoredMNIST and Waterbirds benchmark datasets. As shown in Table 1, SaTeen achieves the SOTA performance, surpassing DeYO (Lee et al., 2024) by 2.9% and 0.8% in average accuracy, and by 14.5% and 1.7% in worst-group accuracy on the two datasets, respectively. The proposed SaTeen benefits from intra-sample structure alignment, effectively enhancing dominant-features learning from reliable samples.

**Mixed Shifts** This is a more realistic and complex scenario

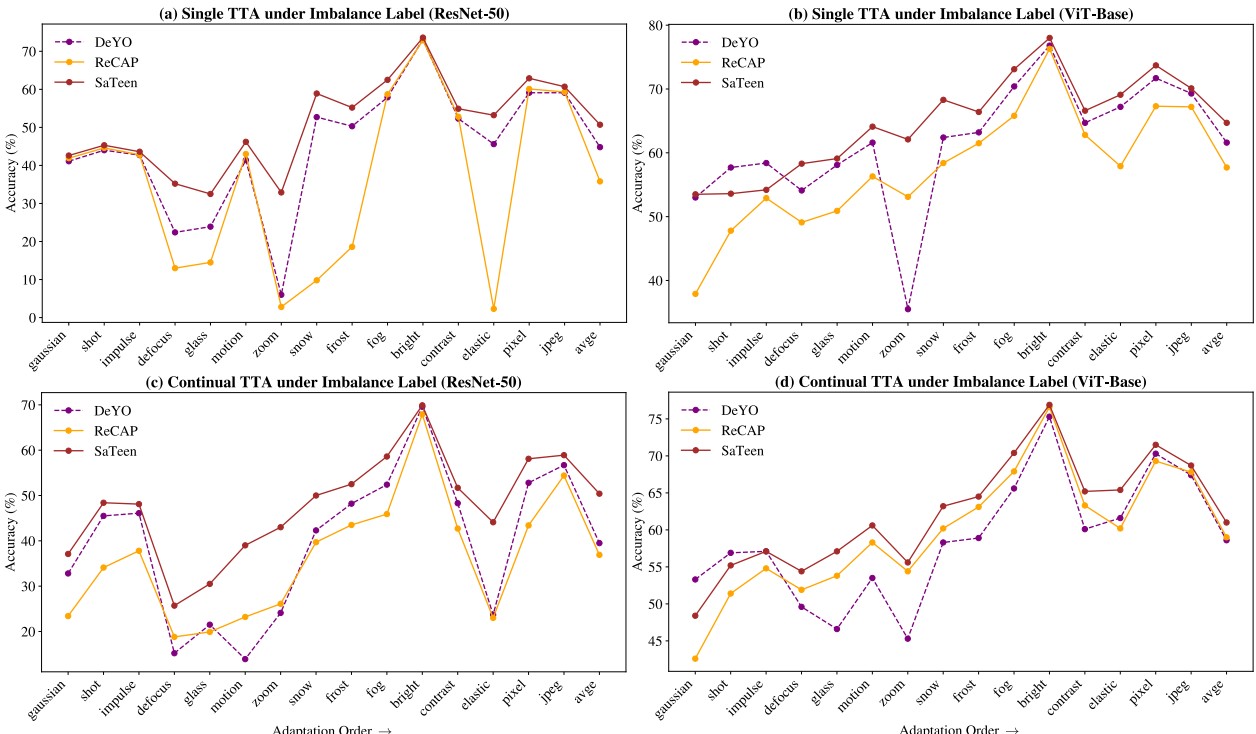

*Figure 5.* Accuracy (%) of single and continual TTA on ImageNet-C across corruption types under imbalance labels at severity 5.

involving mixed shifts, where the model must adapt to a sequence that mixes all 15 ImageNet-C corruption types at varying severity levels. Table 2 reports the average accuracy. We can see that SaTeen achieves the highest accuracy at both severity levels for ResNet50-GN and VitBase-LN backbones, establishing a new SOTA with an average accuracy of 52.1% and 66.3%, respectively. Besides, our method consistently outperforms all strong baselines, including ReCAP (Hu et al., 2025) and DeYO (Lee et al., 2024). These results demonstrate that the structural alignment in SaTeen provides a generalized and robust adaptation strategy, effectively handling the compounded challenge of diverse corruption types and severity levels.

**Single TTA** We evaluate the proposed method on the standard single TTA task, where a model must adapt to a single, persistently shifted target domain. The experiment is conducted using the ImageNet-C benchmark, which applies 15 distinct corruption types spanning noise, blur, weather, and digital categories to the ImageNet validation set, serving as the target domains for adaptation. As shown in Table 3, SaTeen consistently achieves the highest average accuracy across all corruption types for both backbones. Specifically, it surpasses DeYO (Lee et al., 2024) by a significant margin of 7.5% for ResNet50-GN and 1.3% for VitBase-LN. Furthermore, the ablation study within the table demonstrates a clear gain when employing both structural alignments over only intra-sample alignment. Benefiting from the two align-

ments, SaTeen achieves the SOTA performance to handle diverse and challenging in single TTA.

**Continual TTA** This is a more challenging scenario, where the model must sequentially adapt to a stream of different distribution shifts over time. The evaluation is performed on the same ImageNet-C benchmark, with the model adapting to each of the 15 corruption types in sequence. As shown in Table 4, SaTeen establishes a new SOTA performance, achieving the highest average accuracy of 47.1% for ResNet50-GN and 64.6% for VitBase-LN. Our method delivers a significant improvement compared to strong baselines, including DeYO (Lee et al., 2024) and REM (Han et al., 2025). Besides, SaTeen (Intra and Inter) is better than SaTeen (Intra-only), implying that the combination of both alignments leads to more improvements. On the other hand, the inter-sample structure alignment under continual TTA yields larger improvements than under single TTA scenario. It shows IPCA provides effective subspace transformation for continues model adaption.

**Imbalance Label Scenario** Besides, we assess the robustness of SaTeen under imbalance label scenario (Niu et al., 2023), where a large number of samples belonging to the same class arrives in a concentrated manner, through experiments on both single and continual TTA. As shown in Fig. 5, SaTeen maintains consistently strong and stable performance across nearly all corruption types in both adapta-

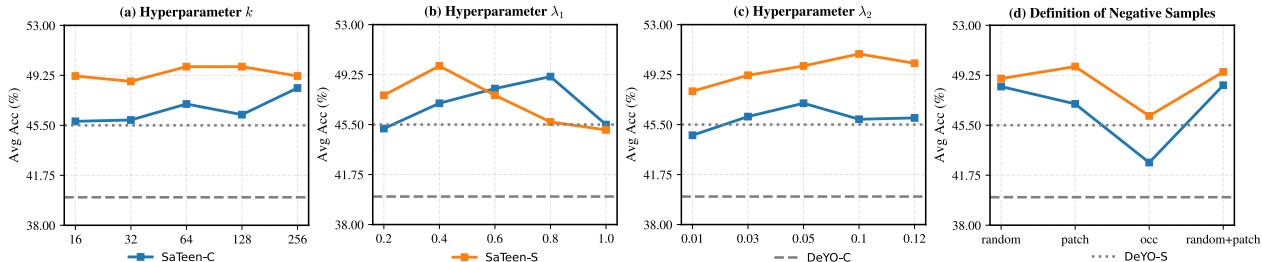

*Figure 6.* Hyperparameter experiments of SaTeen on ImageNet-C under single and continual TTA at severity 5.

*Table 5.* Ablation study of different components on ResNet50-GN under continual TTA. (2)···(15) indicate different cases.

| | $\lambda_1$ | $\lambda_2$ | $\tau$ | $\alpha$ | Avg. |
|---|---|---|---|---|---|
| DeYO | – | – | – | – | 40.1 |
| Tent | | | | | 8.2 |
| (2) | | | ✓ | | 34.2 |
| (3) | | ✓ | | | 23.4 |
| (4) | ✓ | | | | 27.3 |
| (5) | | ✓ | ✓ | | 36.8 |
| (6) | ✓ | | ✓ | | 38.5 |
| (7) | ✓ | ✓ | | | 31.5 |
| (8) | ✓ | ✓ | ✓ | | 42.3 |
| (9) | | | | ✓ | 8.2 |
| (10) | | ✓ | | ✓ | 13.5 |
| (11) | ✓ | | | ✓ | 32.4 |
| (12) | ✓ | | | ✓ | 40.3 |
| (13) | | ✓ | ✓ | ✓ | 37.0 |
| (14) | ✓ | | ✓ | ✓ | 43.2 |
| (15) | ✓ | ✓ | | ✓ | 34.8 |
| **SaTeen** | ✓ | ✓ | ✓ | ✓ | **47.1** |

*Table 6.* Comparison of different methods in terms of the number of parameters on VitBase under continual TTA, time taken for processing, and accuracy (%) on the test set.

| Method | Parameters | Time | Acc |
|---|---|---|---|
| VitBase-LN | 0 | - | 44.2 |
| Tent (Wang et al., 2021) | 0.04M | 8m35s | 49.0 |
| SAR (Niu et al., 2023) | 0.04M | 17m21s | 54.8 |
| DeYO (Lee et al., 2024) | 0.04M | 10m17s | 61.6 |
| ViDA (Liu et al., 2024b) | 93.7M | 54m48s | 56.6 |
| CMAE (Liu et al., 2024a) | 86.5M | 59m56s | 57.5 |
| REM (Han et al., 2025) | 0.03M | 17m21s | 60.8 |
| ReCAP (Hu et al., 2025) | 0.04M | 13m45s | 57.7 |
| **SaTeen (Ours)** | 0.04M | 12m21s | **64.6** |

tion modes, significantly outperforming established baseline methods. Notably, it demonstrates a remarkable ability to preserve high accuracy under extreme shifts and effectively mitigates catastrophic forgetting during sequential adaptation. These findings confirm the exceptional robustness of SaTeen in imbalance label scenarios, attributable to the efficacy of the structural alignment mechanism in managing complex and varied real-world distribution shifts. Detailed results for the imbalanced label setting, severity level 3, and *batch size 1* are provided in the Appendix B.

### 4.3. Hyperparameter and Ablation Study

We conduct hyperparameter sensitivity studies on $k$, $\lambda_1$, $\lambda_2$, and the definition of the negative sample. As shown in Fig. 6, the results indicate that both intra-sample structural alignment and inter-sample structural alignment lead to clear performance improvements, while being insensitive to the choice of hyperparameters. In particular, even the negative sample definition that randomly samples another instance from the current batch outperforms DeYO (Lee et al., 2024) in both single TTA and continual TTA under mild scenarios.

However, this definition becomes ineffective in imbalance label and batch size 1 scenarios.

To assess the contribution of each component, we further conducted ablation studies: Table 5 shows that the intra-sample structural alignment is the most critical component in SaTeen. Notably, even without sample selection, it still outperforms the SOTA DeYO (Lee et al., 2024). Moreover, using intra-sample structural alignment alone achieves performance comparable to using sample selection alone. We also observe that combining intra-sample structural alignment and inter-sample structural alignment further boosts the model performance. More ablation studies are provided in the Appendix C.

### 4.4. Efficiency Comparison

We conduct an efficiency comparison of different methods, as shown in Table 6. In terms of trainable parameters, our SaTeen only requires training 0.04M parameters, which is comparable to Tent (Wang et al., 2021), SAR (Niu et al., 2023), and DeYO (Lee et al., 2024), and is second only to REM (Han et al., 2025). Regarding training time, SaTeen achieves competitive efficiency, being slightly slower than Tent (Wang et al., 2021) and DeYO (Lee et al., 2024). Despite its lightweight design, SaTeen attains the best accuracy among all compared methods.

# 5. Conclusion

In this paper, we propose SaTeen for Test-Time Adaptation (TTA) by combining intra- and inter-sample structural alignments. The theoretical analysis and extensive experimental results demonstrate that SaTeen outperforms existing methods, including ReCAP and DeYO, under both single-domain and continual TTA scenarios. The integration of structural alignment helps the model adapt effectively while can mitigate model collapse and catastrophic forgetting, achieving state-of-the-art results across various benchmarks.

# Acknowledgements

The authors would like to thank the four anonymous reviewers for their valuable comments. This work was supported in part by the National Natural Science Foundation of China under Grant Nos. 62272392, 62537001, 62433016, and U22A2025.

# Impact Statement

This paper presents work whose goal is to advance the field of test-time adaptation and robust machine learning under distribution shifts. There are many potential societal consequences of our work, none of which we feel must be specifically highlighted here.

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

# Appendix

The appendix is organized as follows:

- **Appendix A** contains all missing proofs in the main manuscript.

- **Appendix B** presents additional experimental results on supplementary datasets.

- **Appendix C** provides further ablation studies and visualizations.

- **Appendix D** details the datasets and the methods used for comparison.

- **Appendix E** expands on related work in relevant fields.

## A. Theoretical Proof

**Assumption A.1** (Strong Density Condition). Given the parameters $\mu^-, \mu^+, c_t, c_t^*, r_t > 0$, we assume that the distributions $\mathcal{P}_S$ and $\mathcal{P}_T$ are absolutely continuous with respect to the Lebesgue measure $\lambda[\cdot]$ in Euclidean space. Let $B(\mathbf{x}, r) = \{\mathbf{x}' : \|\mathbf{x}' - \mathbf{x}\| \leq r\}$ denote the closed ball center at point $\mathbf{x}$ with radius $r$. We further assume that $\forall x_t \sim \mathcal{P}_T$ and $r \in (0, r_t]$, the following conditions hold:

$$\lambda[\mathcal{P}_S \cap B(x_t, r)] \geq c_t \lambda[B(x_t, r)]$$

$$\lambda[\mathcal{P}_T \cap B(x_t, r)] \geq c_t^* \lambda[B(x_t, r)]$$

$$\mu^- < \frac{\partial \mathcal{P}_S}{\partial \lambda} < \mu^+; \quad \mu^- < \frac{\partial \mathcal{P}_T}{\partial \lambda} < \mu^+$$

**Assumption A.2** (L-Lipschitz Continuity). Let $f(\cdot; \theta) = g(h(\cdot; \theta))$ be an estimated hypothesis on $H$. We assume that there exists a constant $L$ such that $\forall \mathbf{x}_1, \mathbf{x}_2 \in \mathcal{P}_S \cup \mathcal{P}_T$ and $\forall \theta_1, \theta_2$, the encoder $h(\cdot)$ satisfies the following condition:

$$\|h(\mathbf{x}_1; \theta) - h(\mathbf{x}_2; \theta)\| \leq L\|\mathbf{x}_1 - \mathbf{x}_2\|$$

and

$$\|h(\mathbf{x}; \theta_1) - h(\mathbf{x}; \theta_2)\| \leq L\|\theta_1 - \theta_2\|$$

**Assumption A.3** (Taylor Approximation). Let $f(\cdot; \theta) = g(h(\cdot; \theta))$ be a $L$-Lipschitz Continuous hypothesis on $H$. Let $\mathbf{z} = h(\mathbf{x}; \theta)$ and $f(\mathbf{x}; \theta) = g(\mathbf{z})$. We assume that there exists a constant $r^*$ such that $\forall x_1, x_2 \in \mathcal{P}_S \cup \mathcal{P}_T$, if $\|\mathbf{z}_1 - \mathbf{z}_2\| \leq r^*$, then $f(\mathbf{x}_2; \theta) = g(\mathbf{z}_2)$ can be approximated using the first-order Taylor expansion at $\mathbf{z}_1$ as follows:

$$f(\mathbf{x}_2; \theta) = f(\mathbf{x}_1; \theta) + J_g(\mathbf{z}_1)(\mathbf{z}_2 - \mathbf{z}_1) + o(\|\mathbf{z}_1 - \mathbf{z}_2\|)$$

where $f(\mathbf{x}_1; \theta) = g(\mathbf{z}_1)$, $J_g(\mathbf{z}_1)$ is the Jacobian matrix of $g$ evaluated at $\mathbf{z}_1$, and $o(\|\mathbf{z}_1 - \mathbf{z}_2\|)$ represents the higher-order terms in the expansion.

**Proof of Theorem (2.2).**

**Theorem A.4.** *(**Imbalanced Prediction Induces Model Collapse**) With specific assumptions, if the $\delta$ is sufficiently large, then there exists a parameter-update direction $\mathbf{v} \in \mathbb{R}^{|\Phi|}$, where $\Phi$ denotes the model parameter space. The following properties hold:*

*(1) For a sample $\mathbf{x}$, its likelihood on $c$, i.e, $p_c(\mathbf{x})$, increases monotonically along $\mathbf{v}$;*

*(2) For a sample $\mathbf{x}$, when $p_c$ becomes sufficiently large, the Hessian of the entropy $H(\mathbf{p}(\mathbf{x}))$ attains its minimum eigenvalue along the collapse direction $\mathbf{v}$, namely*

$$\lambda_{\min}\left(\nabla^2 H(\mathbf{p}(\mathbf{x}))\right) = \mathbf{v}^\top \nabla^2 H(\mathbf{p}(\mathbf{x})) \mathbf{v} \leq 0.$$

*(3) On the target-domain distribution, the EM gradient and the parameter update have a positive projection onto $\mathbf{v}$, i.e.,*

$$\mathbb{E}_{\mathbf{x} \sim \mathcal{P}_T(\mathbf{x})}\left[\langle \nabla H(\mathbf{p}(\mathbf{x})), \mathbf{v} \rangle\right] > 0.$$

*Proof.* (1) Let $H(\cdot)$ denote the entropy function. Let $h(\cdot; \theta_h)$ and $g(\cdot)$ denote the encoder and the classifier, respectively. Given an input $x$, let

$$\mathbf{z} = h(\mathbf{x}; \theta_h) \in \mathbb{R}^d$$

be the encoder output, and

$$\mathbf{u} = g(\mathbf{z}) \in \mathbb{R}^{|\mathcal{C}|}$$

be the logits. Let $\mathbf{W}$ denote the weight matrix of $g$, and $\mathbf{W}_c$ the weight vector corresponding to class $c$.

As in the Tent (Wang et al., 2021) setting, we only update the parameters of the normalization layers, we construct a feature direction $\mathbf{v} = \mathbf{v}_h$ such that

$$\mathbf{v}_h = \nabla_{\theta_h} \left( \mathbf{r}^\top \mathbf{z} \right).$$

where $\mathbf{r}$ is a feature direction satisfying $\mathbf{r}^\top \mathbf{z} \neq 0$, that is, for samples containing noisy features. Here, $\mathbf{v}_h$ is the direction in the normalization-layer parameters that amplifies the feature component along $\mathbf{r}$.

This direction $\mathbf{v} = \mathbf{v}_h$ induces a change in logits:

$$\Delta \mathbf{u} = \mathbf{W} \mathbf{r}.$$

By choosing $\mathbf{W_c r} > 0$ and $\mathbf{W_j r} < 0$. Since $d > C$, the direction $\mathbf{r}$ is attainable. We ensure

$$\Delta \mathbf{u}_c > 0, \qquad \Delta \mathbf{u}_j \leq 0 \ \text{ for } j \neq c.$$

Thus, moving along $\mathbf{v}$ increases $u_c$, and hence $p_c = \text{softmax}(\mathbf{u})_c$ monotonically. This proof is inspired by (Chen et al., 2026). $\square$

*Proof.* (2) Let $\mathbf{p} = \text{softmax}(\mathbf{u})$, we have

$$\left[ \nabla_u^2 H \right]_{ij} := \frac{\partial^2 H}{\partial u_i \, \partial u_j} = p_i (\delta_{ij} - p_j)(H - \log p_i - 1).$$

As $p_c \to 1$, let $\mu = 1 - p_c \to 0^+$, and for $p_j \approx \dfrac{\mu}{|\mathcal{C}| - 1}$ for $j \neq c$. Then in the Hessian matrix,

$$\begin{cases} \dfrac{\partial^2 H}{\partial u_c^2} = p_c(1 - p_c)(H - \log p_c - 1), & \text{Diagonal for } c, \\[2mm] \dfrac{\partial^2 H}{\partial u_j^2} = p_j(1 - p_j)(H - \log p_j - 1), & \text{Diagonal for } j \neq c, \\[2mm] \dfrac{\partial^2 H}{\partial u_i \, \partial u_j} = -p_i p_j (H - \log p_i - \log p_j - 1), & \text{Off-diagonal.} \end{cases}$$

we obtain the following approximation:

$$\begin{cases} \dfrac{\partial^2 H}{\partial u_c^2} \approx (1 - \mu)\mu(H + \mu - 1) \approx \mu\left( \mu \log \dfrac{|\mathcal{C}| - 1}{\mu} + \mu - 1 \right) \approx -\mu, \\[3mm] \dfrac{\partial^2 H}{\partial u_j^2} \approx \dfrac{\mu}{|\mathcal{C}| - 1}\left( \mu \log \dfrac{|\mathcal{C}| - 1}{\mu} - \log \dfrac{\mu}{|\mathcal{C}| - 1} - 1 \right) \approx \dfrac{\mu}{|\mathcal{C}| - 1} \log \dfrac{|\mathcal{C}| - 1}{\mu} > 0, \\[3mm] \dfrac{\partial^2 H}{\partial u_i \, \partial u_j} \approx -p_i p_j(-\log p_j) > 0. \end{cases}$$

That is, as $p_c \to 1$, we have

$$\mathbf{v}^\top \nabla_\theta^2 H \, \mathbf{v} \ \to \ \lambda_{\min}(\nabla_\theta^2 H).$$

This proof is inspired by (Jin et al., 2017; Chen et al., 2026). $\square$

*Proof.* (3)

The entropy of the softmax distribution induced by logits $\mathbf{u} = \{u_k\}_{k=1}^C$ is

$$H(\mathbf{p}) = -\sum_{k=1}^C p_k(\mathbf{x}) \log p_k(\mathbf{x}), \qquad p_k(\mathbf{x}) = \frac{e^{u_k}}{\sum_{j=1}^C e^{u_j}}, \qquad \log p_k(\mathbf{x}) = u_k - \log \sum_{j=1}^C e^{u_j}$$

$$= -\sum_{k=1}^C \frac{e^{u_k}}{\sum_{j=1}^C e^{u_j}} u_k + \log \sum_{j=1}^C e^{u_j} = -\frac{\sum_{k=1}^C e^{u_k} u_k}{\sum_{j=1}^C e^{u_j}} + \log \sum_{j=1}^C e^{u_j}.$$

Taking the derivative of $H(\mathbf{p}(\mathbf{x}))$ with respect to $u_i$, we obtain

$$\frac{\partial H(\mathbf{p}(\mathbf{x}))}{\partial u_i} = -\frac{\partial}{\partial u_i} \left( \frac{\sum_{k=1}^C e^{u_k} u_k}{\sum_{j=1}^C e^{u_j}} \right) + \frac{\partial}{\partial u_i} \left( \log \sum_{j=1}^C e^{u_j} \right)$$

$$= -\frac{\left(e^{u_i} u_i + e^{u_i}\right) \sum_{j=1}^C e^{u_j} - \left(\sum_{k=1}^C e^{u_k} u_k\right) e^{u_i}}{\left(\sum_{j=1}^C e^{u_j}\right)^2} + \frac{e^{u_i}}{\sum_{j=1}^C e^{u_j}}$$

$$= -\frac{e^{u_i} \left( u_i \sum_{j=1}^C e^{u_j} - \sum_{k=1}^C e^{u_k} u_k \right)}{\left(\sum_{j=1}^C e^{u_j}\right)^2} = -\frac{e^{u_i} \sum_{j=1}^C e^{u_j} \left(u_i - u_j\right)}{\left(\sum_{j=1}^C e^{u_j}\right)^2}, \qquad i = 1, 2, \ldots, C.$$

The entropy has a negative gradient with respect to the largest logit, a positive gradient with respect to the smallest logit, while the sign for the remaining logits is generally indeterminate. From (1), we know that $u_c$ is monotonic along the direction $\mathbf{v}$. Therefore, when the predicted class of $\mathbf{x}$ is $c$, we have $\left\langle \frac{\partial H(\mathbf{x})}{\partial \theta}, \mathbf{v} \right\rangle > 0$. When the imbalance predicate rate $\delta$ is sufficiently large, we assume that the collapse direction is consistent across all samples, i.e., aligned with $\mathbf{v}$. We partition the target-domain distribution into two sets, $A$ and $B$, where $A$ contains samples whose predicted class is $c$, and $B$ contains samples whose predicted class is not $c$. Thus, we have:

$$\mathbb{E}_{\mathbf{x} \sim \mathcal{P}_T(\mathbf{x})} \left[\langle \nabla H(\mathbf{p}(\mathbf{x})), \mathbf{v} \rangle\right] = \delta \, \mathbb{E}_{\mathbf{x} \in A} \left[\langle \nabla H(\mathbf{p}(\mathbf{x})), \mathbf{v} \rangle\right] + (1 - \delta) \, \mathbb{E}_{\mathbf{x} \in B} \left[\langle \nabla H(\mathbf{p}(\mathbf{x})), \mathbf{v} \rangle\right]$$

Taking the derivative of $H(\mathbf{x})$ with respect to $\theta$ and applying the chain rule, we obtain

$$\nabla_\theta H(\mathbf{x}) = \left( \frac{\partial \mathbf{h}(\mathbf{x}; \theta)}{\partial \theta} \right) \mathbf{W}^T \nabla_{\mathbf{u}} H(\mathbf{x}).$$

By Assumption A.2, $h(\mathbf{x}; \theta)$ is $L$-Lipschitz in $\theta$, and $\mathbf{W}$ is constant, which implies $\|\nabla_\theta h(\mathbf{x}; \theta)\|_{\mathrm{op}} \leq L$. Moreover,

$$\left\|\nabla_{\mathbf{u}} H(\mathbf{x})\right\|_2 \leq \sqrt{C} \left( \log C + 2 + \frac{1}{e} \right).$$

then, we have

$$\left\|\nabla_\theta H(\mathbf{x})\right\|_2 \leq L \|\mathbf{W}\|_{\mathrm{op}} \sqrt{C} \left( \log C + 2 + \frac{1}{e} \right).$$

Therefore, $\left\|\nabla_\theta H(\mathbf{x})\right\|_2$ is bounded. Consequently, its projection onto any direction $\mathbf{v}$ is also bounded. We denote an upper bound by

$$|\langle \nabla_\theta H(\mathbf{x}), \mathbf{v} \rangle| \leq M$$

Moreover, we assume

$$\mathbb{E}_{\mathbf{x} \in A}[\langle \nabla H(\mathbf{p}(\mathbf{x})), \mathbf{v} \rangle] > m_+ > 0, \qquad \mathbb{E}_{\mathbf{x} \in B}[\langle \nabla H(\mathbf{p}(\mathbf{x})), \mathbf{v} \rangle] > -M,$$

where $m_+$ indicates that the expected directional derivative over set $A$ is bounded away from zero. Let $\delta \triangleq \mathbb{P}(\mathbf{x} \in A)$ so that $\mathbb{P}(\mathbf{x} \in B) = 1 - \delta$. Then

$$\mathbb{E}_{\mathbf{x} \in \mathcal{P}_T}[\langle \nabla H(\mathbf{p}(\mathbf{x})), \mathbf{v}\rangle] = \delta\, \mathbb{E}_{\mathbf{x} \in A}[\langle \nabla H(\mathbf{p}(\mathbf{x})), \mathbf{v}\rangle] + (1-\delta)\, \mathbb{E}_{\mathbf{x} \in B}[\langle \nabla H(\mathbf{p}(\mathbf{x})), \mathbf{v}\rangle] > \delta m_+ - (1-\delta)M.$$

Hence, whenever

$$\delta > \frac{M}{M + m_+},$$

Theorem (2.2)(3) holds. □

**Proof of Theorem (3.1).**

**Theorem A.5.** *(The $\mathcal{L}_1(\lambda_1, \mathbf{x})$ Loss Mitigates Model Collapse) For a batch of samples $\{\mathbf{x}_i\}_{i=1}^{N_{\mathrm{bs}}}$ that the imbalance prediction rate is $\delta$, even though $\delta$ is large, if $\lambda_1$ is sufficiently large, then after a $\mathcal{L}_1(\lambda_1, \mathbf{x})$-loss gradient update, the parameter update has a non-positive projection onto $\mathbf{v}$, i.e.,*

$$\mathbb{E}_{\mathbf{x} \sim \mathcal{P}_T(\mathbf{x})}[\langle \nabla \mathcal{L}_1(\lambda_1, (\mathbf{x})), \mathbf{v}\rangle] \leq 0.$$

*Proof*: The following gives the derivative of the entropy $H(\mathbf{x})$ with respect to $u_i$:

$$\frac{\partial \mathrm{CE}(sg[\mathbf{p}(\tilde{\mathbf{x}})] \,\|\, \mathbf{p}(\mathbf{x}))}{\partial u_i} = \frac{\partial\left(-\sum p_i(\tilde{\mathbf{x}}) \log p_i(\mathbf{x})\right)}{\partial u_i}$$
$$= -\frac{\partial \sum_i p(\tilde{\mathbf{x}})\left(u_i - \log \sum_j e^{u_j}\right)}{\partial u_i}$$
$$= -\sum (p_i(\tilde{\mathbf{x}}) - p_j(\tilde{\mathbf{x}}))\, p_i(\mathbf{x})$$

As can be seen, when $\tilde{\mathbf{x}}$ is also predicted as class $c$, the derivative of $\mathrm{CE}(sg[\mathbf{p}(\tilde{\mathbf{x}})] \,\|\, \mathbf{p}(\mathbf{x}))$ w.r.t. the target logit $u_c$ is negative. Hence, along the direction $\mathbf{v}$ we have

$$\left\langle \nabla_\theta\left(-\mathrm{CE}(sg[\mathbf{p}(\tilde{\mathbf{x}})] \,\|\, \mathbf{p}(\mathbf{x}))\right), \mathbf{v}\right\rangle < 0.$$

Since $\tilde{\mathbf{x}}$ is the structure-disrupted counterpart augmentation of $\mathbf{x}$ and retains only noisy features, we further assume

$$\mathbb{E}_{\mathbf{x} \sim \mathcal{P}_T}\left[\left\langle \nabla_\theta\left(-\mathrm{CE}(sg[\mathbf{p}(\tilde{\mathbf{x}})] \,\|\, \mathbf{p}(\mathbf{x}))\right), \mathbf{v}\right\rangle\right] < m_- < 0,$$

where $m_-$ indicates that the expected directional derivative over $\mathcal{P}_T$ is bounded away from zero (in the negative direction).

Therefore, the expected directional derivative of $\mathcal{L}_1$ over the target distribution $\mathcal{P}_T$ satisfies

$$\mathbb{E}_{\mathbf{x} \sim \mathcal{P}_T}[\langle \nabla_\theta \mathcal{L}_1(\lambda_1; \mathbf{x}), \mathbf{v}\rangle] = \delta\, \mathbb{E}_{\mathbf{x} \in A}[\langle \nabla_\theta H(\mathbf{p}(\mathbf{x})), \mathbf{v}\rangle] + (1-\delta)\, \mathbb{E}_{\mathbf{x} \in B}[\langle \nabla_\theta H(\mathbf{p}(\mathbf{x})), \mathbf{v}\rangle] - \lambda_1\, \mathbb{E}_{\mathbf{x} \sim \mathcal{P}_T}[\langle \nabla_\theta H(\mathbf{p}(\mathbf{x})), \mathbf{v}\rangle]$$
$$\leq \delta\, m_+ - (1-\delta)\, M + \lambda_1\, m_-.$$

Hence, whenever

$$\lambda_1 > -\frac{\delta\, m_+ - (1-\delta)\, M}{m_-}$$

Theorem (3.1) holds.

**Proof of Theorem (3.2)**

**Lemma A.6** (Upper-Bounded Source-Domain Covariance Estimated from Reliable Target Samples). *Let $\Omega := \bigcup_{\mathbf{x} \sim \mathcal{P}_T} \mathcal{B}(\mathbf{x}, r^*)$ be the set of balls near the test data, where $\mathcal{B}(\mathbf{x}, r^*) = \{\mathbf{x}_0 : \|\mathbf{x}_0 - \mathbf{x}\| \leq r^*\}$. We sample $N_t$ source samples from $\mathcal{P}_S \cap \Omega$ and $N_t$ target samples from $\mathcal{P}_T$. Let $\mathbf{V}_S$ and $\mathbf{V}_T$ denote the covariance matrices. When Assumptions A.1, A.2, and A.3 hold, with probability at least $1 - \epsilon_p$, we have:*

$$\|\mathbf{V}_S - \mathbf{V}_T\|_F \leq \epsilon_{\mathbf{v}}. \tag{19}$$

*Here, $\epsilon_{\mathbf{v}}$ denotes an upper bound on the source-domain variance estimated from reliable target samples, which is related to the prediction confidence and the prediction error of the sampled instances.*

Lemma A.6 is taken from (You et al., 2025).

**Lemma A.7** (Davis-Kahan). *For real symmetric matrices, given spectral gap $\widetilde{\delta}$, we have*

$$\|\sin\Theta(\mathbf{U}_S,\mathbf{U}_T)\|_F \le \frac{\|\mathbf{V}_S - \mathbf{V}_T\|_F}{\widetilde{\delta}},$$

*where $\mathbf{U}_S$ and $\mathbf{U}_T$ are the orthogonal bases.*

Combining Lemma A.6 and Lemma A.7, we have:

$$\|\sin\Theta(\mathbf{U}_S,\mathbf{U}_T)\|_F \le \epsilon_{\mathbf{u}} = \frac{\epsilon_{\mathbf{v}}}{\widetilde{\delta}}, \tag{20}$$

which yields Theorem 3.2.

**Why** $\mathrm{CE}(sg[\mathbf{p}(\tilde{\mathbf{x}})] \,\|\, \mathbf{p}(\mathbf{x}))$ **not** $\mathrm{CE}(sg[\mathbf{p}(\mathbf{x})] \,\|\, \mathbf{p}(\tilde{\mathbf{x}}))$   The following gives the derivative of the entropy $H(\mathbf{x})$ with respect to $\theta$:

$$\frac{\partial H(\mathbf{x})}{\partial\theta} = \frac{\partial\left(-\sum p_i(\mathbf{x})\log p_i(\mathbf{x})\right)}{\partial\theta}$$
$$= -\sum\left(\log p_i(\mathbf{x}) + 1\right)\frac{\partial p_i(\mathbf{x})}{\partial\theta}.$$

The following gives the derivative of the cross-entropy $\mathrm{CE}(sg[\mathbf{p}(\tilde{\mathbf{x}})] \,\|\, \mathbf{p}(\mathbf{x}))$ with respect to $\theta$:

$$\frac{\partial\mathrm{CE}(sg[\mathbf{p}(\tilde{\mathbf{x}})] \,\|\, \mathbf{p}(\mathbf{x}))}{\partial\theta} = \frac{\partial\left(-\sum p_i(\tilde{\mathbf{x}})\log p_i(\mathbf{x})\right)}{\partial\theta}$$
$$= -\sum\left(\frac{p_i(\tilde{\mathbf{x}})}{p_i(\mathbf{x})}\frac{\partial p_i(\mathbf{x})}{\partial\theta}\right),$$

where the $\frac{p_i(\tilde{\mathbf{x}})}{p_i(\mathbf{x})} > 0$.

The following gives the derivative of the cross-entropy $\mathrm{CE}(\mathbf{p}(\mathbf{x}) \,\|\, sg[\mathbf{p}(\tilde{\mathbf{x}})])$ with respect to $\theta$:

$$\frac{\partial\mathrm{CE}(\mathbf{p}(\mathbf{x}) \,\|\, sg[\mathbf{p}(\tilde{\mathbf{x}})])}{\partial\theta} = \frac{\partial\left(-\sum p_i(\mathbf{x})\log p_i(\tilde{\mathbf{x}})\right)}{\partial\theta}$$
$$= -\sum\left(\log p_i(\tilde{\mathbf{x}})\frac{\partial p_i(\mathbf{x})}{\partial\theta}\right),$$

where the $\log p_i(\tilde{\mathbf{x}}) < 0$.

When we mainly focus on the influence of $\frac{\partial p_c(\mathbf{x})}{\partial\theta}$ for reliable samples, *i.e.*, low-entropy samples where $c$ is the predicted class, we have $1 + \log p_c(\mathbf{x}) > 0$. In this case, the gradient direction of $\mathrm{CE}(sg[\mathbf{p}(\tilde{\mathbf{x}})] \,\|\, \mathbf{p}(\mathbf{x}))$ is consistent with that of $H(\mathbf{x})$. Therefore, applying a negative sign suppresses the EM gradient contributed by this term, but $\mathrm{CE}(sg[\mathbf{p}(\mathbf{x})] \,\|\, \mathbf{p}(\tilde{\mathbf{x}}))$ cannot.

**Derivation of Entropy Decouple, Eq. (2).**

*Proof.* By definition, the predictive entropy is

$$H(\mathbf{p}(\mathbf{x})) = -\sum_{i=1}^{C} p_i(\mathbf{x})\log p_i(\mathbf{x})$$
$$= -\sum_{i=1}^{C} p_i(\mathbf{x})\left(u_i - \log\sum_{j=1}^{C} e^{u_j}\right)$$
$$= -\sum_{i=1}^{C} p_i(\mathbf{x})\,u_i + \log\sum_{j=1}^{C} e^{u_j}$$

This is exactly Eq. (2)                                                                                        $\square$.

**Derivation of $\mathcal{L}_1(\lambda_1, \mathbf{x})$ Decouple, Eq. (7).**

*Proof.* By definition, the loss in (6) is

$$\mathcal{L}_1(\lambda_1, \mathbf{x}) = -\sum_{i=1}^{C} p_i(\mathbf{x}) \log p_i(\mathbf{x}) - \lambda_1 \left( -\sum_{i=1}^{C} p_i(\tilde{\mathbf{x}}) \log p_i(\mathbf{x}) \right)$$

$$= -\sum_{i=1}^{C} \left( p_i(\mathbf{x}) - \lambda_1 p_i(\tilde{\mathbf{x}}) \right) \log p_i(\mathbf{x})$$

$$= -\sum_{i=1}^{C} \left( p_i(\mathbf{x}) - \lambda_1 p_i(\tilde{\mathbf{x}}) \right) \left( u_i - \log \sum_{j=1}^{C} e^{u_j} \right)$$

$$= -\sum_{i=1}^{C} \left( p_i(\mathbf{x}) - \lambda_1 p_i(\tilde{\mathbf{x}}) \right) u_i + (1 - \lambda_1) \log \sum_{j=1}^{C} e^{u_j}.$$

which matches Eq. (7). □

## B. Further Experiments

### B.1. Wild TTA

*Table 7.* Accuracy (%) of Single TTA on ImageNet-C across corruption types under **label shifts** at severity level 5.

| Wild TTA | Noise | | | Blur | | | | Weather | | | | Digital | | | | Avg. |
|---|---|---|---|---|---|---|---|---|---|---|---|---|---|---|---|---|
| | Gauss. | Shot | Impl. | Defoc. | Glass | Motion | Zoom | Snow | Frost | Fog | Brit. | Contr. | Elastic | Pixel | JPEG | |
| ResNet-50-GN | 17.9 | 19.7 | 17.8 | 19.9 | 11.4 | 21.4 | 24.7 | 40.3 | 47.2 | 33.4 | 69.2 | 36.3 | 18.6 | 28.3 | 52.2 | 31.0 |
| Tent (Wang et al., 2021) | 16.2 | 5.5 | 0.9 | 12.6 | 8.1 | 10.2 | 7.5 | 15.0 | 4.4 | 1.8 | 16.6 | 2.5 | 0.2 | 1.7 | 8.2 | 8.2 |
| SAR (Niu et al., 2023) | 34.6 | 36.5 | 35.9 | 18.9 | 21.5 | 33.5 | **32.9** | 27.5 | 44.7 | 49.6 | 71.6 | 46.8 | 6.8 | 51.9 | 56.2 | 37.9 |
| DeYO (Lee et al., 2024) | 41.1 | 44.0 | 42.7 | 22.4 | 23.9 | 41.4 | 6.0 | 52.7 | 50.3 | 57.9 | 73.1 | 52.3 | 45.6 | 59.1 | 59.1 | 44.8 |
| ReCAP (Hu et al., 2025) | 41.9 | 44.6 | 42.8 | 13.0 | 14.5 | 43.0 | 2.8 | 9.8 | 18.6 | 58.7 | 72.9 | 52.8 | 2.3 | 60.1 | 59.3 | 35.8 |
| **SaTeen (Ours)** | **42.6** | **45.3** | **43.6** | **35.2** | **32.5** | **46.2** | **32.9** | **58.9** | **55.2** | **62.5** | **73.6** | **54.9** | **53.2** | **62.9** | **60.7** | **50.7** |
| VitBase-LN | 9.5 | 6.8 | 8.2 | 29.0 | 23.4 | 33.9 | 27.1 | 15.9 | 26.5 | 47.2 | 54.6 | 44.1 | 30.5 | 44.5 | 47.8 | 28.5 |
| Tent (Wang et al., 2021) | 38.0 | 1.0 | 36.7 | 54.4 | 52.1 | 58.0 | 51.8 | 11.3 | 11.3 | 69.1 | 76.0 | 65.9 | 56.2 | 69.1 | 66.2 | 47.8 |
| SAR (Niu et al., 2023) | 50.7 | 56.2 | 55.1 | 41.8 | 39.1 | 53.9 | 48.2 | 58.7 | 55.9 | 58.2 | 76.2 | 42.8 | 50.1 | 67.1 | 67.3 | 54.8 |
| DeYO (Lee et al., 2024) | 53.0 | **57.7** | **58.4** | 54.1 | 58.1 | 61.6 | 35.5 | 62.4 | 63.2 | 70.4 | 76.8 | 64.7 | 67.2 | 71.7 | **69.3** | 61.6 |
| ReCAP (Hu et al., 2025) | 37.9 | 47.8 | 52.9 | 49.1 | 50.9 | 56.3 | 53.1 | 58.4 | 61.5 | 65.8 | 76.3 | 62.8 | 57.9 | 67.3 | 67.2 | 57.7 |
| **SaTeen (Ours)** | **53.5** | 53.6 | 54.2 | **58.3** | **59.1** | **64.1** | **62.1** | **68.3** | **66.4** | **73.1** | **78.0** | **66.6** | **69.1** | **73.7** | 70.1 | **64.7** |

We evaluate the robustness of adaptation methods under a challenging Wild Scenario with significant label shifts. Table 7 presents the wild TTA experimental results for a source model pre-trained on ImageNet, using each corruption on ImageNet-C as the target domain. The model sequentially adapts to the target domains over time, and we compare the average error for each domain. SaTeen consistently outperforms all baseline methods across all 15 corruption types in terms of accuracy, demonstrating the effectiveness of our approach. This advantage holds across both ResNet-50 and Vision Transformer backbones. Compared with the current state-of-the-art method for mild continual test-time adaptation, DeYO (Lee et al., 2024), SaTeen achieves a $5.9\%$ absolute accuracy improvement under the ResNet-50 backbone and a $3.1\%$ improvement under the Vision Transformer backbone.

We evaluate continual test-time adaptation under a wild scenario with severe label shifts, assessing the model's ability to sequentially adapt to multiple different corruptions over time. The experiment is conducted on ImageNet-C at severity level 5, and the detailed accuracy for each of the 15 corruption types alongside the final average is presented in Table 8. The results show that SaTeen achieves the best overall performance, with an average accuracy of 50.4% for ResNet-50 and 61.0% for Vision Transformer. It demonstrates a significant improvement of 10.9% over the prior best method DeYO (Lee et al., 2024) with ResNet-50 and maintains a consistent advantage across most corruption types, particularly in challenging ones like defocus and motion blur. This confirms that the dual structural alignment in SaTeen effectively prevents error accumulation and enables robust long-term adaptation in highly unstable real-world environments.

We examine the performance of adaptation methods under an extreme yet practical condition. The experiment is conducted on the ImageNet-C benchmark at severity level 5, testing Single TTA performance across all 15 corruption types when only

*Table 8.* Accuracy (%) of Continual TTA on ImageNet-C across corruption types under **label shifts** at severity level 5.

| Wild CTTA | Noise | | | Blur | | | | Weather | | | | Digital | | | | Avg. |
|---|---|---|---|---|---|---|---|---|---|---|---|---|---|---|---|---|
| | Gauss. | Shot | Impl. | Defoc. | Glass | Motion | Zoom | Snow | Frost | Fog | Brit. | Contr. | Elastic | Pixel | JPEG | |
| ResNet50-GN | 17.9 | 19.7 | 17.8 | 19.9 | 11.4 | 21.4 | 24.7 | 40.3 | 47.2 | 33.4 | 69.2 | 36.3 | 18.6 | 28.3 | 52.2 | 31.0 |
| Tent (Wang et al., 2021) | 14.9 | 3.4 | 0.6 | 12.1 | 6.5 | 6.1 | 4.2 | 8.4 | 1.7 | 0.9 | 9.4 | 1.2 | 0.1 | 0.8 | 5.7 | 5.1 |
| SAR (Niu et al., 2023) | 22.8 | 33.6 | 37.5 | 18.3 | 19.9 | 21.3 | 24.0 | 38.3 | 42.8 | 43.2 | 66.4 | 40.3 | 23.5 | 40.9 | 52.8 | 35.0 |
| DeYO (Lee et al., 2024) | 32.8 | 45.5 | 46.1 | 15.2 | 21.5 | 13.9 | 24.1 | 42.3 | 48.2 | 52.4 | 69.6 | 48.3 | 23.7 | 52.8 | 56.7 | 39.5 |
| ReCAP (Hu et al., 2025) | 23.4 | 34.1 | 37.8 | 18.8 | 19.9 | 23.2 | 26.1 | 39.7 | 43.5 | 45.9 | 67.9 | 42.7 | 23.0 | 43.4 | 54.4 | 36.9 |
| **SaTeen (Ours)** | **37.1** | **48.4** | **48.1** | **25.7** | **30.5** | **39.0** | **43.0** | **50.0** | **52.5** | **58.6** | **69.9** | **51.7** | **44.1** | **58.1** | **58.9** | **50.4** |
| VitBase-LN | 9.5 | 6.8 | 8.2 | 29.0 | 23.4 | 33.9 | 27.1 | 15.9 | 26.5 | 47.2 | 54.6 | 44.1 | 30.5 | 44.5 | 47.8 | 28.5 |
| Tent (Wang et al., 2021) | 27.4 | 38.9 | 46.4 | 41.8 | 43.6 | 51.9 | 47.8 | 53.3 | 57.3 | 61.5 | 74.9 | 59.0 | 50.8 | 63.5 | 64.7 | 51.5 |
| SAR (Niu et al., 2023) | 48.2 | 53.8 | 56.2 | 53.4 | 55.6 | 59.2 | 55.9 | 61.4 | 63.4 | 68.8 | 76.4 | 63.6 | 62.1 | 69.9 | 68.1 | 60.8 |
| DeYO (Lee et al., 2024) | 53.3 | 56.9 | 57.1 | 49.6 | 46.6 | 53.5 | 45.3 | 58.3 | 58.9 | 65.6 | 75.3 | 60.1 | 61.6 | 70.3 | 67.4 | 58.6 |
| ReCAP (Hu et al., 2025) | 42.6 | 51.4 | 54.8 | 51.9 | 53.8 | 58.3 | 54.4 | 60.2 | 63.1 | 67.9 | 76.6 | 63.3 | 60.2 | 69.3 | 67.8 | 59.0 |
| **SaTeen (Ours)** | 48.4 | 55.2 | 57.1 | 54.4 | 57.1 | 60.6 | 55.6 | 63.2 | 64.5 | 70.4 | 76.9 | 65.2 | 65.4 | 71.5 | 68.7 | 61.0 |

*Table 9.* Accuracy (%) under Single TTA on ImageNet-C across corruption types under **batch size 1** at severity level 5.

| Single TTA | Noise | | | Blur | | | | Weather | | | | Digital | | | | Avg. |
|---|---|---|---|---|---|---|---|---|---|---|---|---|---|---|---|---|
| | Gauss. | Shot | Impl. | Defoc. | Glass | Motion | Zoom | Snow | Frost | Fog | Brit. | Contr. | Elastic | Pixel | JPEG | |
| ResNet-50-GN | 18.0 | 19.8 | 17.9 | 19.8 | 11.4 | 21.4 | 24.9 | 40.4 | 47.3 | 33.6 | 69.3 | 36.3 | 18.6 | 28.4 | 52.3 | 30.6 |
| Tent (Wang et al., 2021) | 2.5 | 2.9 | 2.5 | 13.5 | 3.6 | 18.6 | 17.6 | 15.3 | 23.0 | 1.4 | 70.4 | 42.2 | 6.2 | 49.2 | 53.8 | 21.5 |
| EATA (Niu et al., 2022) | 24.9 | 28.0 | 25.8 | 18.3 | 17.0 | 31.2 | 29.8 | 42.5 | 44.1 | 41.3 | 70.9 | 44.2 | 27.6 | 46.8 | 55.4 | 36.5 |
| SAR (Niu et al., 2023) | 25.5 | 28.0 | 24.9 | 18.7 | 16.3 | 28.6 | 31.4 | 46.2 | 44.9 | 33.4 | 72.8 | 44.3 | 15.3 | 47.1 | 56.1 | 35.6 |
| DeYO (Lee et al., 2024) | 41.2 | 44.3 | 42.5 | 22.4 | 24.7 | 41.8 | 21.9 | 54.8 | 51.6 | 21.9 | 73.1 | 53.2 | 48.5 | 59.8 | 59.6 | 44.1 |
| ReCAP (Hu et al., 2025) | 41.6 | 44.8 | 42.3 | 9.5 | 19.6 | 44.3 | 46.2 | 7.6 | 53.1 | 35.1 | 73.1 | 53.6 | 23.6 | 60.7 | 60.5 | 41.0 |
| **SaTeen (Ours)** | **41.7** | **45.2** | **44.3** | **32.3** | **29.7** | 44.0 | 36.7 | **57.4** | **54.4** | 30.2 | **73.9** | **55.3** | **50.2** | **61.0** | **61.1** | **47.8** |

one sample is available per adaptation step. The detailed accuracy for each corruption and the final average are presented in Table 9. Our analysis shows that SaTeen achieves the highest average accuracy of 47.8%, outperforming all baseline methods. While performance varies across different corruption types, SaTeen demonstrates superior robustness, especially in challenging categories like defocus, glass, and motion blur. These results confirm that the dual structural alignment in SaTeen remains highly effective even under severe data efficiency constraints, making it suitable for real-world scenarios where adaptation must occur sequentially with minimal memory footprint.

*Table 10.* Accuracy (%) of Single TTA on ImageNet-C across corruption types under **label shifts** at severity level 3.

| Wild Single TTA | Noise | | | Blur | | | | Weather | | | | Digital | | | | Avg. |
|---|---|---|---|---|---|---|---|---|---|---|---|---|---|---|---|---|
| | Gauss. | Shot | Impl. | Defoc. | Glass | Motion | Zoom | Snow | Frost | Fog | Brit. | Contr. | Elastic | Pixel | JPEG | |
| ResNet50-GN | 54.5 | 52.9 | 53.1 | 44.1 | 21.1 | 49.5 | 39.0 | 54.9 | 54.0 | 55.8 | 75.3 | 69.8 | 59.6 | 59.5 | 66.2 | 53.3 |
| Tent (Wang et al., 2021) | 58.8 | 58.6 | 58.7 | 37.7 | 27.3 | 54.8 | 42.9 | 51.1 | 41.0 | 62.4 | 75.2 | 70.2 | 63.0 | 63.6 | 66.4 | 56.1 |
| SAR (Niu et al., 2023) | 60.3 | 60.0 | 60.1 | 45.9 | 37.5 | 58.0 | 49.3 | 57.8 | 53.2 | 65.1 | 76.2 | 71.0 | 67.0 | 65.6 | 67.3 | 59.7 |
| DeYO (Lee et al., 2024) | 63.5 | 63.7 | 63.3 | 54.7 | 44.9 | 62.3 | 55.0 | 61.7 | 58.2 | 69.2 | 76.7 | 73.1 | 71.0 | 69.8 | 69.5 | 63.1 |
| ReCAP (Hu et al., 2025) | 63.7 | 63.8 | 63.2 | 55.0 | 45.2 | 62.5 | 56.0 | 61.6 | 58.0 | 69.1 | 76.8 | 73.0 | 70.9 | 70.0 | 69.5 | 63.5 |
| **SaTeen (Ours)** | **64.6** | **64.6** | **64.1** | **56.5** | **51.2** | **64.7** | **58.4** | **64.4** | **60.6** | **71.2** | **77.1** | **74.0** | **72.4** | **71.6** | **70.9** | **65.7** |
| VitBase-LN | 51.5 | 46.8 | 50.3 | 48.6 | 37.1 | 54.6 | 41.3 | 35.2 | 33.4 | 67.5 | 69.0 | 74.7 | 65.8 | 65.9 | 63.5 | 52.7 |
| Tent (Wang et al., 2021) | 68.7 | 68.0 | 68.0 | 67.8 | 64.1 | 70.7 | 63.6 | 67.8 | 38.4 | 76.3 | 78.7 | 79.4 | 75.9 | 76.8 | 73.6 | 68.4 |
| SAR (Niu et al., 2023) | 68.7 | 68.1 | 68.2 | 68.0 | 64.5 | 70.8 | 63.9 | 68.0 | 64.1 | 76.4 | 78.8 | 79.4 | 76.1 | 76.9 | 73.7 | 70.4 |
| DeYO (Lee et al., 2024) | **71.7** | **71.7** | **71.4** | **70.7** | **68.9** | **73.8** | 68.8 | 72.6 | 69.9 | 77.6 | **80.3** | 79.6 | **78.2** | **79.0** | 76.7 | **73.1** |
| ReCAP (Hu et al., 2025) | 71.5 | 71.3 | 71.3 | 70.5 | 68.6 | 73.3 | 68.6 | 72.0 | 69.7 | **77.9** | 80.2 | **79.8** | 78.0 | 78.8 | 76.5 | 73.0 |
| **SaTeen (Ours)** | 71.3 | 71.4 | 71.2 | 69.8 | 68.8 | 73.5 | **69.2** | **72.9** | **70.2** | 77.3 | 80.2 | 79.3 | 78.0 | 78.7 | **76.7** | **73.2** |

We evaluate single test-time adaptation under label shifts at severity level 3 to test the method's performance across a range of shift intensities. The experiment is conducted on the ImageNet-C benchmark, and the performance across all corruption types is detailed in Table 10, which reports the final average accuracy for each method. The results show that SaTeen achieves the highest average accuracy of 65.7% with ResNet-50, demonstrating clear improvements over all baselines. With Vision Transformer, SaTeen attains a competitive average accuracy of 73.2%, matching or slightly exceeding the performance of other top methods. These findings indicate that the dual structural alignment in SaTeen provides strong and reliable adaptation under moderate distribution shifts, with its advantage being particularly pronounced for convolutional network architectures.

We assess continual test-time adaptation under label shifts at severity level 3 to measure the method's stability under moderate sequential distribution changes. The experiment is conducted on the ImageNet-C benchmark, and the results are summarized in Table 11, which reports the average accuracy across all 15 corruption types. The findings show that SaTeen

*Table 11.* Accuracy (%) of Continual TTA on ImageNet-C across corruption types under **label shifts** at severity level 3.

| Wild CTTA | Noise | | | Blur | | | | Weather | | | | Digital | | | | Avg. |
|---|---|---|---|---|---|---|---|---|---|---|---|---|---|---|---|---|
| | Gauss. | Shot | Impl. | Defoc. | Glass | Motion | Zoom | Snow | Frost | Fog | Brit. | Contr. | Elastic | Pixel | JPEG | |
| ResNet-50-GN | 54.5 | 52.9 | 53.1 | 44.1 | 21.1 | 49.5 | 39.0 | 54.9 | 54.0 | 55.8 | 75.3 | 69.8 | 59.6 | 59.5 | 66.2 | 53.3 |
| Tent (Wang et al., 2021) | 58.2 | 59.2 | 58.7 | 40.7 | 17.1 | 39.5 | 20.1 | 24.2 | 16.7 | 17.3 | 34.5 | 21.6 | 9.1 | 5.3 | 6.7 | 30.5 |
| SAR (Niu et al., 2023) | 57.9 | 61.6 | 63.1 | 47.4 | 33.1 | 53.1 | 48.2 | 51.5 | 51.9 | 63.2 | 75.3 | 72.4 | 64.3 | 63.0 | 68.9 | 58.3 |
| DeYO (Lee et al., 2024) | 62.2 | 65.2 | 64.3 | 50.2 | 44.2 | 59.2 | 54.8 | 57.8 | 57.7 | 66.8 | 74.8 | 71.9 | 69.1 | 68.2 | 69.5 | 61.7 |
| ReCAP (Hu et al., 2025) | 62.4 | 65.4 | 64.5 | 49.8 | 43.7 | 59.3 | 56.3 | 57.8 | 58.0 | 66.8 | 74.8 | 71.7 | 69.2 | 68.2 | 69.5 | 61.8 |
| **SaTeen (Ours)** | **63.9** | **65.9** | **65.2** | **54.5** | **51.5** | **62.8** | **57.3** | **61.5** | **59.5** | **68.5** | 74.8 | **72.1** | **71.0** | **70.3** | **70.1** | **64.9** |
| VitBase-LN | 51.5 | 46.8 | 50.3 | 48.6 | 37.1 | 54.6 | 41.3 | 35.2 | 33.4 | 67.5 | 69.0 | 74.7 | 65.8 | 65.9 | 63.5 | 52.7 |
| Tent (Wang et al., 2021) | 60.5 | 64.7 | 67.5 | 60.7 | 57.5 | 69.1 | 60.2 | 62.3 | 63.0 | 73.3 | 79.2 | 77.9 | 76.1 | 75.5 | 74.8 | 68.2 |
| SAR (Niu et al., 2023) | 60.1 | 64.6 | 67.4 | 61.1 | 58.0 | 69.5 | 60.7 | 62.8 | 63.5 | 73.7 | 79.7 | 78.4 | 76.6 | 77.0 | 75.4 | 68.5 |
| DeYO (Lee et al., 2024) | 69.9 | 70.4 | 67.7 | 63.2 | 63.4 | 69.6 | 60.5 | 58.5 | 64.8 | 70.3 | 76.4 | 74.6 | 56.8 | 73.3 | 70.9 | 67.4 |
| ReCAP (Hu et al., 2025) | 66.6 | 70.5 | 71.5 | 67.0 | 65.2 | 72.3 | 65.2 | 67.6 | 67.3 | 74.5 | 80.0 | 78.6 | 76.8 | 77.4 | 76.5 | 71.8 |
| **SaTeen (Ours)** | **71.3** | **72.0** | **71.7** | **68.4** | **68.2** | 71.8 | **68.1** | **70.8** | **68.4** | **75.1** | 78.6 | 77.3 | 76.6 | 77.3 | 75.6 | **72.8** |

obtains the highest average accuracy of 64.9% with ResNet-50 and 72.8% with Vision Transformer. It consistently achieves better overall performance compared to all baselines, demonstrating effective and stable adaptation even when the model must adjust sequentially to multiple different corruptions over time. These results reinforce that the dual structural alignment in SaTeen is a reliable strategy for continual adaptation across varying intensities of distribution shift.

## B.2. Mild Experiment at Severity 3

*Table 12.* Accuracy (%) under Single TTA on ImageNet-C across corruption types at severity level 3.

| Single TTA | Noise | | | Blur | | | | Weather | | | | Digital | | | | Avg. |
|---|---|---|---|---|---|---|---|---|---|---|---|---|---|---|---|---|
| | Gauss. | Shot | Impl. | Defoc. | Glass | Motion | Zoom | Snow | Frost | Fog | Brit. | Contr. | Elastic | Pixel | JPEG | |
| ResNet50-GN | 54.5 | 52.9 | 53.1 | 44.4 | 21.2 | 49.8 | 39.3 | 54.9 | 54.1 | 55.8 | 75.3 | 69.7 | 59.6 | 59.7 | 66.4 | 54.1 |
| Tent (Wang et al., 2021) | 59.1 | 58.6 | 58.3 | 39.0 | 27.9 | 54.7 | 41.1 | 51.3 | 41.4 | 62.0 | 75.2 | 70.1 | 62.3 | 63.7 | 66.4 | 55.4 |
| EATA (Niu et al., 2022) | 52.3 | 52.9 | 51.7 | 35.7 | 30.1 | 46.4 | 39.6 | 43.8 | 39.8 | 55.7 | 72.4 | 66.6 | 54.7 | 56.0 | 56.2 | 50.3 |
| SAR (Niu et al., 2023) | 60.8 | 60.5 | 60.2 | 47.9 | 36.7 | 58.2 | 49.7 | 57.9 | 53.6 | 65.0 | 76.4 | 71.0 | 67.0 | 65.8 | 67.6 | 59.9 |
| DeYO (Lee et al., 2024) | 64.0 | 63.9 | 63.2 | 54.0 | 44.9 | 62.2 | 55.1 | 62.1 | 57.9 | 69.2 | 76.9 | 73.2 | 71.2 | 70.2 | 69.8 | 63.8 |
| ReCAP (Hu et al., 2025) | 63.1 | 62.8 | 62.4 | 53.4 | 41.9 | 60.9 | 53.6 | 60.0 | 56.2 | 68.0 | 76.9 | 72.7 | 69.8 | 68.3 | 68.7 | 62.5 |
| **SaTeen (Ours)** | **64.6** | **64.6** | **64.1** | **56.5** | **51.2** | **64.7** | **58.4** | **64.4** | **60.6** | **71.2** | **77.1** | **74.0** | **72.4** | **71.6** | **70.9** | **65.7** |
| VitBase-LN | 51.5 | 46.8 | 50.4 | 48.7 | 37.1 | 54.7 | 41.6 | 35.1 | 33.3 | 68.0 | 69.3 | 74.9 | 65.9 | 66.0 | 63.6 | 53.8 |
| Tent (Wang et al., 2021) | 68.7 | 68.0 | 68.1 | 68.2 | 63.8 | 70.9 | 63.8 | 67.6 | 41.9 | 76.3 | 78.8 | 79.5 | 75.9 | 76.7 | 73.7 | 69.5 |
| EATA (Niu et al., 2022) | 65.3 | 62.6 | 63.6 | 63.0 | 57.1 | 66.3 | 59.3 | 64.5 | 61.0 | 73.3 | 76.9 | 75.9 | 74.2 | 74.8 | 73.1 | 67.4 |
| SAR (Niu et al., 2023) | 68.8 | 68.2 | 68.4 | 68.3 | 64.7 | 71.0 | 64.2 | 68.1 | 66.0 | 76.4 | 79.0 | 79.6 | 76.2 | 77.1 | 74.1 | 71.3 |
| DeYO (Lee et al., 2024) | 71.7 | 71.6 | 71.4 | 70.6 | 68.9 | 73.9 | 69.2 | 72.4 | 69.7 | 77.9 | 80.2 | 79.5 | 78.2 | 78.7 | 76.7 | 74.0 |
| ReCAP (Hu et al., 2025) | 71.0 | 70.6 | 70.6 | 70.1 | 67.7 | 73.0 | 67.4 | 70.9 | 68.7 | 77.8 | 80.1 | 80.2 | 77.8 | 78.5 | 76.3 | 73.9 |
| **SaTeen (Ours)** | **72.7** | **72.6** | **72.5** | **71.4** | **70.1** | **74.5** | **70.1** | **73.3** | **70.8** | **78.4** | **80.9** | **80.4** | **78.8** | **79.4** | **77.5** | **74.9** |

We evaluate the proposed method on the standard Single Test-Time Adaptation task at severity level 3, where a model must adapt to a single, persistently shifted target domain. The experiment is conducted using the ImageNet-C benchmark, which applies 15 distinct corruption types spanning noise, blur, weather, and digital categories to the ImageNet validation set, serving as the target domain for adaptation. As shown in Table 12, SaTeen consistently achieves the highest average accuracy across all corruption types for both backbones. Specifically, it surpasses the previous state-of-the-art method, DeYO (Lee et al., 2024), by a significant margin of 4.3% for ResNet-50 and 1.3% for Vision Transformer. Furthermore, the ablation study within the table demonstrates a clear performance gain when employing the complete dual-alignment framework over using only intra-sample alignment. The synergistic integration of intra-sample and inter-sample structure alignment, as instantiated in SaTeen, provides a robust and generalized adaptation strategy that effectively handles diverse and challenging distribution shifts encountered in Single TTA.

We examine the more challenging Continual Test-Time Adaptation setting severity level 3, where a model must sequentially adapt to a stream of different distribution shifts over time. The evaluation is performed on the same ImageNet-C benchmark, with the model adapting to each of the 15 corruption types in sequence. As shown in Table 13, SaTeen establishes a new state-of-the-art, achieving the highest average accuracy of 47.1% for ResNet-50 and 64.6% for Vision Transformer. The method demonstrates a significant advantage over strong baselines including DeYO (Lee et al., 2024) and REM (Han et al., 2025). Notably, the performance improvement conferred by the complete dual-alignment framework, compared to using only intra-sample alignment, is more substantial in this sequential setting, especially for the ResNet-50 backbone. This demonstrates that the inter-sample structure alignment, which leverages a stable feature subspace built from reliable samples, is critically effective in mitigating error accumulation and catastrophic forgetting during continual adaptation to

*Table 13.* Accuracy (%) of Continual TTA on ImageNet-C across corruption types at severity level 3.

| Continual TTA | Noise | | | Blur | | | | Weather | | | | Digital | | | | Avg. |
|---|---|---|---|---|---|---|---|---|---|---|---|---|---|---|---|---|
| | Gauss. | Shot | Impl. | Defoc. | Glass | Motion | Zoom | Snow | Frost | Fog | Brit. | Contr. | Elastic | Pixel | JPEG | |
| ResNet50-GN | 54.5 | 52.9 | 53.1 | 44.4 | 21.2 | 49.8 | 39.3 | 54.9 | 54.1 | 55.8 | 75.3 | 69.7 | 59.6 | 59.7 | 66.4 | 54.1 |
| Tent (Wang et al., 2021) | 58.5 | 58.9 | 58.3 | 40.0 | 15.7 | 33.5 | 13.5 | 16.0 | 9.2 | 7.5 | 17.1 | 7.6 | 2.2 | 2.0 | 2.6 | 25.5 |
| SAR (Niu et al., 2023) | 58.0 | 62.0 | 63.4 | 45.4 | 34.4 | 52.2 | 48.6 | 52.7 | 53.0 | 63.6 | 75.4 | 72.6 | 64.5 | 63.5 | 69.1 | 58.9 |
| DeYO (Lee et al., 2024) | 60.8 | 64.8 | 64.6 | 48.8 | 37.3 | 57.6 | 51.7 | 55.0 | 57.0 | 65.3 | 75.3 | 72.8 | 68.0 | 67.0 | 69.7 | 60.4 |
| ReCAP (Hu et al., 2025) | 60.6 | 64.6 | 64.6 | 48.3 | 36.6 | 57.7 | 53.3 | 55.5 | 57.2 | 65.5 | 75.1 | 72.5 | 67.8 | 66.7 | 69.6 | 60.7 |
| **SaTeen (Ours)** | **62.7** | **66.0** | **65.4** | **52.9** | **47.6** | **61.3** | **56.6** | **61.0** | **60.0** | **67.8** | **75.5** | **72.8** | **70.4** | **69.5** | **70.7** | **64.0** |
| VitBase-LN | 51.5 | 46.8 | 50.4 | 48.7 | 37.1 | 54.7 | 41.6 | 35.1 | 33.3 | 68.0 | 69.3 | 74.9 | 65.9 | 66.0 | 63.6 | 53.8 |
| Tent (Wang et al., 2021) | 57.8 | 61.4 | 64.9 | 58.2 | 53.8 | 67.8 | 58.4 | 59.5 | 60.7 | 73.5 | 79.2 | 77.9 | 75.8 | 76.4 | 74.2 | 67.3 |
| SAR (Niu et al., 2023) | 57.4 | 60.9 | 64.6 | 58.3 | 53.7 | 68.0 | 58.5 | 59.7 | 60.9 | 74.0 | 79.4 | 78.3 | 76.0 | 76.7 | 74.6 | 66.8 |
| DeYO (Lee et al., 2024) | 65.3 | 69.9 | 71.4 | 66.0 | 64.3 | 72.3 | 64.8 | 67.3 | 67.2 | 74.8 | 80.4 | 79.0 | 77.1 | 77.7 | 76.5 | 71.6 |
| ReCAP (Hu et al., 2025) | 63.8 | 68.5 | 70.5 | 65.5 | 62.5 | 71.6 | 63.6 | 66.1 | 66.1 | 74.0 | 80.1 | 78.8 | 76.6 | 77.2 | 76.3 | 71.5 |
| **SaTeen (Ours)** | **72.7** | **72.6** | **72.5** | **71.4** | **70.1** | **74.5** | **70.1** | **73.3** | **70.8** | **78.4** | **80.9** | **80.4** | **78.8** | **79.4** | **77.5** | **74.9** |

non-stationary data streams.

## B.3. ImageNet-R and VisDA-2021

We conduct additional experiments using the ImageNet-R (Hendrycks et al., 2021) and VisDA-2021 (Bashkirova et al., 2022) datasets with ResNet and ViT architectures. These datasets are characterized by diverse distributions.

*Table 14.* Comparisons with state-of-the-art methods on ImageNet-R under mild & imbalanced label single TTA regarding Accuracy (%). We report mean±std over 3 independent runs. The best results are in bold.

| Model+Method | Mild | | | Imbalanced Label | | | Average |
|---|---|---|---|---|---|---|---|
| | ResNet50-GN | VitBase-LN | Avg | ResNet50-GN | VitBase-LN | Avg | |
| Noadapt Model | 40.8 | 43.1 | 42.0 | 40.8 | 43.1 | 42.0 | 42.0 |
| Tent (Wang et al., 2021) | 42.1 | 42.6 | 42.4 | 42.2 | 46.7 | 44.5 | 43.5 |
| EATA (Niu et al., 2022) | 43.7 | 52.3 | 48.0 | 42.0 | 50.3 | 46.3 | 47.3 |
| SAR (Niu et al., 2023) | 45.2 | 54.1 | 49.2 | 44.4 | 54.4 | 49.4 | 49.3 |
| DeYO (Lee et al., 2024) | 46.1 | 58.2 | 52.2 | 46.7 | 58.5 | 52.6 | 52.4 |
| ReCAP (Hu et al., 2025) | 50.1 | 59.8 | 55.5 | 49.3 | 60.1 | 54.7 | 55.1 |
| SaTeen (Ours) | **51.2**±0.4 | **60.9**±0.1 | **56.1**±0.3 | **50.2**±0.2 | **61.1**±0.2 | **55.2**±0.2 | **55.7**±0.3 |

*Table 15.* Comparisons with state-of-the-art methods on VisDA-2021 under mild & imbalanced label single TTA regarding Accuracy (%). We report mean±std over 3 independent runs. The best results are in bold.

| Model+Method | Mild | | | Imbalanced Label | | | Average |
|---|---|---|---|---|---|---|---|
| | ResNet50-GN | VitBase-LN | Avg | ResNet50-GN | VitBase-LN | Avg | |
| Noadapt Model | 43.5 | 44.3 | 43.9 | 43.5 | 44.3 | 43.9 | 43.9 |
| Tent (Wang et al., 2021) | 43.6 | 50.6 | 47.1 | 43.7 | 50.1 | 46.9 | 47.0 |
| EATA (Niu et al., 2022) | 44.0 | 49.3 | 46.7 | 43.5 | 51.6 | 47.6 | 47.0 |
| SAR (Niu et al., 2023) | 44.7 | 52.1 | 48.3 | 44.8 | 54.1 | 49.5 | 48.9 |
| DeYO (Lee et al., 2024) | 44.8 | 57.5 | 51.2 | 45.2 | 57.1 | 51.2 | 51.2 |
| ReCAP (Hu et al., 2025) | 46.2 | 59.7 | 53.0 | 48.6 | **59.6** | 54.1 | 53.6 |
| SaTeen (Ours) | **48.3**±0.7 | **59.7**±0.2 | **54**±0.5 | **49.8**±0.5 | 59.3±0.2 | **54.6**±0.4 | **54.3**±0.5 |

For both datasets, we use a mixed-domain test setup that combines mild shifts with imbalanced label distributions. This stricter protocol is closer to practical deployment settings and provides a stronger stress test of robustness. All experiments follow the same implementation details described in the main paper.

Table 14 summarizes the ImageNet-R results under these shifts. The conclusions mirror those on ImageNet-C: SaTeen achieves the strongest performance across multiple architectures and evaluation scenarios.

We also assess SaTeen on VisDA-2021 and benchmark it against prior state-of-the-art test-time adaptation (TTA) methods.

As reported in Table 15, SaTeen attains the best accuracy in every setting, consistent with the overall trends on the ImageNet benchmarks.

Beyond aggregate results, we examine SaTeen under a broader range of distribution changes. As discussed in the main paper, SaTeen uses an efficient approximation of region-level confidence to reduce prediction inconsistency and to improve optimization stability. Relative to other sample-selection-based TTA baselines, SaTeen yields steady improvements across architectures and shift types, with average gains of $+0.6\%$ on ImageNet-R and $+0.7\%$ on VisDA-2021. Together, these findings highlight the robustness and transferability of our approach in realistic test-time conditions.

### B.4. Scene Classification

*Table 16.* Accuracy (%) under Single TTA on Places365-C across corruption types at severity level 5.

| Single TTA | Noise | | | Blur | | | | Weather | | | | Digital | | | | Avg. |
|---|---|---|---|---|---|---|---|---|---|---|---|---|---|---|---|---|
| | Gauss. | Shot | Impl. | Defoc. | Glass | Motion | Zoom | Snow | Frost | Fog | Brit. | Contr. | Elastic | Pixel | JPEG | |
| ResNet-50-BN | 19.2 | 20.7 | 18.7 | 24.2 | 20.6 | 30.9 | 38.1 | 26.3 | 25.2 | 42.2 | 46.2 | 28.6 | 38.2 | 42.8 | 38.6 | 30.7 |
| Tent (Wang et al., 2021) | 27.0 | 28.5 | 26.2 | 31.0 | 25.8 | 35.9 | 40.5 | 30.2 | 28.1 | 44.9 | 47.1 | 34.8 | 41.2 | 46.0 | 42.4 | 35.3 |
| EATA (Niu et al., 2022) | 30.9 | 32.0 | 30.3 | 34.1 | 29.2 | 39.0 | 42.0 | 34.1 | 31.0 | 45.7 | 47.6 | 38.5 | 42.7 | 46.8 | 43.6 | 37.8 |
| SAR (Niu et al., 2023) | 27.2 | 28.8 | 26.2 | 30.9 | 25.7 | 35.9 | 40.3 | 30.5 | 28.2 | 44.7 | 47.2 | 35.5 | 41.0 | 45.8 | 42.4 | 35.3 |
| DeYO (Lee et al., 2024) | 31.0 | 32.2 | 30.1 | 34.1 | 29.0 | 38.9 | 41.8 | 33.9 | 30.6 | 45.9 | 47.7 | 38.8 | 42.5 | 46.9 | 43.7 | 37.8 |
| **SaTeen (Ours)** | 31.0 | 32.3 | 30.2 | 34.3 | 29.0 | 38.8 | 41.8 | 33.9 | 30.7 | 46.0 | 47.8 | 38.8 | 42.5 | 46.9 | 43.7 | 37.8 |

*Table 17.* Accuracy (%) under Continual TTA on Places365-C across corruption types at severity level 5.

| Continual TTA | Noise | | | Blur | | | | Weather | | | | Digital | | | | Avg. |
|---|---|---|---|---|---|---|---|---|---|---|---|---|---|---|---|---|
| | Gauss. | Shot | Impl. | Defoc. | Glass | Motion | Zoom | Snow | Frost | Fog | Brit. | Contr. | Elastic | Pixel | JPEG | |
| ResNet-50-BN | 19.2 | 20.7 | 18.7 | 24.2 | 20.6 | 30.9 | 38.1 | 26.3 | 25.2 | 42.2 | 46.2 | 28.6 | 38.2 | 42.8 | 38.6 | 30.7 |
| Tent (Wang et al., 2021) | 23.7 | 30.1 | 29.5 | 28.4 | 27.1 | 33.0 | 37.2 | 27.8 | 26.7 | 39.0 | 42.7 | 30.5 | 37.1 | 41.6 | 38.4 | 32.8 |
| EATA (Niu et al., 2022) | 28.6 | 34.0 | 33.2 | 31.8 | 30.6 | 37.0 | 40.6 | 32.3 | 30.7 | 43.4 | 46.2 | 37.4 | 41.5 | 45.6 | 42.5 | 37.0 |
| SAR (Niu et al., 2023) | 23.4 | 30.4 | 30.1 | 28.6 | 27.9 | 34.0 | 38.0 | 29.0 | 28.1 | 41.0 | 45.0 | 33.7 | 39.9 | 44.4 | 41.5 | 34.3 |
| DeYO (Lee et al., 2024) | 27.7 | 34.1 | 33.4 | 29.6 | 29.3 | 34.4 | 37.8 | 29.6 | 28.7 | 40.8 | 44.4 | 36.2 | 39.9 | 44.1 | 41.7 | 35.5 |
| **SaTeen (Ours)** | 27.9 | 34.3 | 33.6 | 30.3 | 29.6 | 34.9 | 38.0 | 30.1 | 29.4 | 41.0 | 44.4 | 36.6 | 39.5 | 44.2 | 41.6 | 35.7 |

*Table 18.* Imbalance prediction rate ($\delta$) on Places365-C across corruption types at severity level 5.

| Imbalance prediction rate(%) | Gauss. | Shot | Impl. | Defoc. | Glass | Motion | Zoom | Snow | Frost | Fog | Brit. | Contr. | Elastic | Pixel | JPEG |
|---|---|---|---|---|---|---|---|---|---|---|---|---|---|---|---|
| ResNet-50-GN | 1.1 | 1.0 | 1.2 | 0.8 | 1.0 | 0.7 | 0.6 | 0.5 | 1.1 | 1.0 | 0.5 | 0.5 | 1.0 | 0.6 | 0.6 |

In addition to the image object classification task, we also conduct experiments on a scene classification task. We utilize Places365 as the training set and ResNet50-BN as the pre-trained model, constructing Places365-C as the test set for the TTA experiments. All experimental configurations are kept consistent with those used for ImageNet-C. As shown in Tables 16 and 17, it can be observed that under the Single TTA setting, EATA, DeYO, and SaTeen all achieve optimal performance. However, in the Continual TTA setting, EATA achieves the best results, while SaTeen only ranks second. Upon analyzing the imbalance prediction rate, as shown in Table 18, we find that in the scene classification task, these corruptions do not introduce structural noise that causes the model to exhibit an extreme bias toward predicting a specific class. Consequently, under these circumstances, SaTeen's strategy of utilizing negative samples to suppress the model's tendency to concentrate predictions on a single class has adverse side effects.

## C. Additional Ablation Study

### C.1. Visualization

From Tables 19 and 20, we observe that the proposed components provide clear complementary benefits. Under single TTA, the Tent baseline achieves an average of $30.6$, while introducing the thresholding module $\tau$ yields consistent gains (e.g., $32.5$ in (2)). In contrast, enabling $\lambda_1$ alone provides only a modest improvement ($31.0$ in (4)). As components are combined, performance increases substantially: $\lambda_1$ with $\tau$ reaches $34.4$ (6), and activating $\lambda_1$, $\lambda_2$, and $\tau$ together further boosts the result to $39.7$ (8), indicating strong synergy between the loss terms and the selection mechanism. Adding $\alpha$ can amplify this effect in certain configurations, e.g., $42.5$ in (12) and $47.1$ in (14). When all four components are enabled, SaTeen attains the best performance of $49.9$, validating the necessity of the full design.

*Table 19.* Ablation study of different components on ResNet50-GN under single TTA.

|  | $\lambda_1$ | $\lambda_2$ | $\tau$ | $\alpha$ | Avg. |
|---|---|---|---|---|---|
| Tent |  |  |  |  | 30.6 |
| (2) |  |  | ✓ |  | 32.5 |
| (3) |  | ✓ |  |  | 30.9 |
| (4) | ✓ |  |  |  | 31.0 |
| (5) |  | ✓ | ✓ |  | 32.0 |
| (6) | ✓ |  | ✓ |  | 34.4 |
| (7) | ✓ | ✓ |  |  | 39.5 |
| (8) | ✓ | ✓ | ✓ |  | 39.7 |
| (9) |  |  |  | ✓ | 30.7 |
| (10) |  |  | ✓ | ✓ | 34.0 |
| (11) |  | ✓ |  | ✓ | 27.7 |
| (12) | ✓ |  |  | ✓ | 42.5 |
| (13) |  | ✓ | ✓ | ✓ | 34.5 |
| (14) | ✓ |  | ✓ | ✓ | 47.1 |
| (15) | ✓ | ✓ |  | ✓ | 42.2 |
| **SaTeen** | ✓ | ✓ | ✓ | ✓ | **49.9** |

*Table 20.* Ablation study of different components on ResNet50-GN under continual TTA. (2)$\cdots$(15) indicate different cases.

|  | $\lambda_1$ | $\lambda_2$ | $\tau$ | $\alpha$ | Avg. |
|---|---|---|---|---|---|
| Tent |  |  |  |  | 8.2 |
| (2) |  |  | ✓ |  | 34.2 |
| (3) |  | ✓ |  |  | 23.4 |
| (4) | ✓ |  |  |  | 27.3 |
| (5) |  | ✓ | ✓ |  | 36.8 |
| (6) | ✓ |  | ✓ |  | 38.5 |
| (7) | ✓ | ✓ |  |  | 31.5 |
| (8) | ✓ | ✓ | ✓ |  | 42.3 |
| (9) |  |  |  | ✓ | 8.2 |
| (10) |  |  | ✓ | ✓ | 13.5 |
| (11) |  | ✓ |  | ✓ | 32.4 |
| (12) | ✓ |  |  | ✓ | 40.3 |
| (13) |  | ✓ | ✓ | ✓ | 37.0 |
| (14) | ✓ |  | ✓ | ✓ | 43.2 |
| (15) | ✓ | ✓ |  | ✓ | 34.8 |
| **SaTeen** | ✓ | ✓ | ✓ | ✓ | **47.1** |

The continual TTA setting is more challenging: Tent drops to $8.2$, yet SaTeen still improves dramatically to 47.1, demonstrating strong robustness under sustained distribution shifts. Similar to the single-TTA case, partial variants are less stable, whereas combining $\lambda_1$, $\tau$, and $\alpha$ already yields strong results (e.g., $40.3$ in (12) and $43.2$ in (14)). Incorporating $\lambda_2$ on top of these components leads to the overall best performance (SaTeen, **47.1**). Overall, the ablations confirm that $\tau$ and $\alpha$ are key to stable improvements, while $\lambda_1/\lambda_2$ contribute additional gains and work synergistically with them across both evaluation protocols.

## D. More Implementation Details

### D.1. Baseline

**Tent** (Wang et al., 2021). We follow all hyper-parameters that are set in Tent unless it does not provide. Specifically, we use SGD as the update rule, with a momentum of 0.9, batch size of 64 (except for the experiments of batch size = 1 and effects of small test batch sizes), and for single TTA and biased scenario, the learning rate of 0.00025/0.001 for ResNet/ViT models. The learning rate for batch size = 1 is set to (0.00025/32) for ResNet models. For continual TTA, the learning rate of 0.0001/0.001 for ResNet/ViT models. The trainable parameters are all affine parameters of normalization layers.

**EATA** (Niu et al., 2022). We follow all hyper-parameters that are set in EATA unless it does not provide. Specifically, the

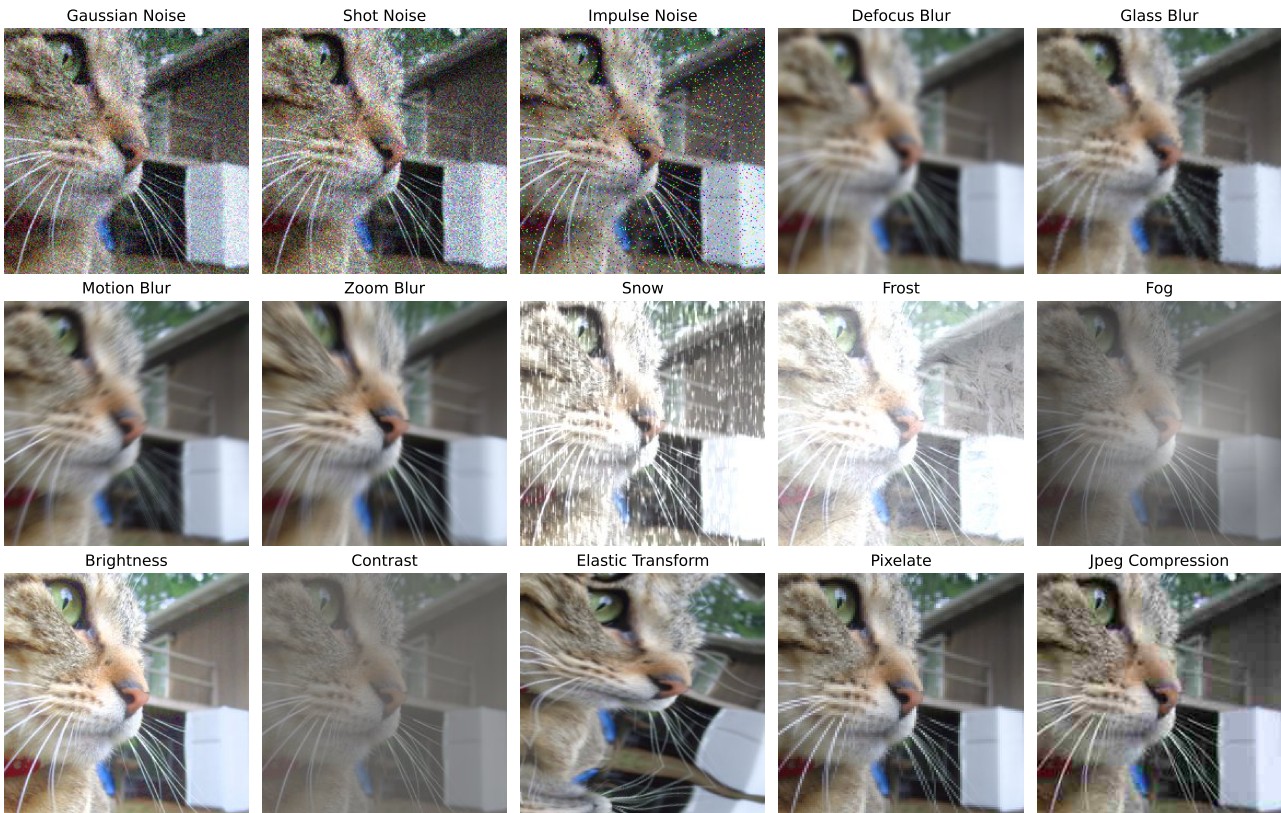

*Figure 7.* Visualizations of different corruption types in ImageNet-C benchmark at severity 2.

entropy constant $E_0$ (for reliable sample identification) is set to $0.1 \times \log 1000$. The $\epsilon$ for redundant sample identification is set to 0.05. The trade-off parameter $\beta$ for entropy loss and regularization loss is set to 2,000. The number of pre-collected in-distribution test samples for Fisher importance calculation is 2,000. The update rule is SGD, with a momentum of 0.9, batch size of 64 (except for the experiments of batch size = 1 and effects of small test batch sizes), and for single TTA and biased scenario, the learning rate of 0.00025/0.001 for ResNet/ViT models. The learning rate for batch size = 1 is set to (0.00025/32) for ResNet models. For continual TTA, the learning rate of 0.0001/0.001 for ResNet/ViT models. The trainable parameters are all affine parameters of normalization layers.

**SAR** (Niu et al., 2023). We follow all hyper-parameters that are set in SAR unless it does not provide. Specifically, the threshold $E_0$ is set to $0.4 \times \log 1000$, the $\rho$ is set to 0.05. The update rule is SGD, with a momentum of 0.9, batch size of 64 (except for the experiments of batch size = 1 and effects of small test batch sizes), and for single TTA and biased scenario, the learning rate of 0.00025/0.001 for ResNet/ViT models. The learning rate for batch size = 1 is set to (0.00025/32) for ResNet models. For continual TTA, the learning rate of 0.0001/0.001 for ResNet/ViT models. The trainable parameters are all affine parameters of normalization layers.

**DeYO** (Lee et al., 2024). We follow all hyper-parameters that are set in DeYO unless it does not provide. Specifically, The required hyperparameters for DeYO are $\tau_{\text{Ent}}, \tau_{\text{PLPD}}$, and $\text{Ent}_0$. We set $\text{Ent}_0$ and $\tau_{\text{Ent}}$ to $0.4 \times \log C$ and $0.5 \times \log C$, respectively. ColoredMNIST and Waterbirds are binary benchmarks, and since they mostly produce one-hot predictions due to their simplicity, we use 0.5 for a threshold of label flip. We do not use entropy filtering and set $\text{Ent}_0 = \log C$ for ColoredMNIST. The update rule is SGD, with a momentum of 0.9, batch size of 64 (except for the experiments of batch size = 1 and effects of small test batch sizes), and for single TTA and biased scenario, the learning rate of 0.00025/0.001 for ResNet/ViT models. The learning rate for batch size = 1 is set to (0.00025/32) for ResNet models. For continual TTA, the learning rate of 0.0001/0.001 for ResNet/ViT models. The trainable parameters are all affine parameters of normalization layers.

**ReCAP** (Hu et al., 2025). We follow all hyper-parameters that are set in ReCAP unless it does not provide. Specifically, the

$\tau$ is set to $0.8 \times \log C$, the $L_0$ is set to 0.8, the reweight threshold is set to 3.0 for ResNet50-GN, the $\tau$ is set to 1.5, the $L_0$ is set to 1.5, the reweight threshold is set to 1.5 for VitBase50-LN. The update rule is SGD, with a momentum of 0.9, batch size of 64 (except for the experiments of batch size = 1 and effects of small test batch sizes), and for single TTA and biased scenario, the learning rate of 0.00025/0.001 for ResNet/ViT models. The learning rate for batch size = 1 is set to (0.00025/32) for ResNet models. For continual TTA, the learning rate of 0.0001/0.001 for ResNet/ViT models. The trainable parameters are all affine parameters of normalization layers.

**SaTeen**. We follow all hyper-parameters that the $\tau$ is set to $0.4 \times \log C$, the $\lambda_1$ is set to 0.4, the $\lambda_2$ is set to 0.05, the $k$ is set to 64. the $f$ is set to 1. The update rule is SGD, with a momentum of 0.9, batch size of 64 (except for the experiments of batch size = 1 and effects of small test batch sizes), and for single TTA and biased scenario, the learning rate of 0.00025/0.001 for ResNet/ViT models. The learning rate for batch size = 1 is set to (0.00025/32) for ResNet models. For continual TTA, the learning rate of 0.0001/0.001 for ResNet/ViT models. The trainable parameters are all affine parameters of normalization layers.

### D.2. Dataset

In this paper, we primarily evaluate the out-of-distribution (OOD) generalization capability of all methods using the widely recognized ImageNet-C, ImageNet-R, VisDA-2021, ColoredMNIST and WaterBirds benchmark (Hendrycks & Dietterich, 2019).

**ImageNet-C** (Hendrycks & Dietterich, 2019) extends the original ImageNet test set (Deng et al., 2009) by introducing a variety of distortions to assess model robustness under real-world distribution shifts. It includes 15 types of corruptions, such as Gaussian noise, defocus blur, motion blur, snow, frost, fog, and various forms of compression and pixelation. Each corruption is assigned five severity levels, with higher levels indicating more substantial perturbations. These distortions simulate real-world degradation, making ImageNet-C a crucial tool for evaluating model performance in challenging scenarios. As shown in Fig. 7, the distortions span a wide range, testing models' ability to adapt to diverse image degradation.

**ImageNet-R** (Hendrycks et al., 2021) consists of 30,000 images representing artistic renditions of 200 ImageNet classes. These images feature creative transformations like paintings, drawings, and sculptures, sourced from platforms such as Flickr and curated through Amazon MTurk annotators. The variations in visual style, texture, and color distribution pose significant challenges, differing substantially from the original ImageNet images.

**VisDA-2021** (Bashkirova et al., 2022), on the other hand, offers a more extensive range of domain shifts. It incorporates images from diverse sources, including ImageNet-O/R/C.

**ColoredMNIST** is a modified version of the classic MNIST dataset, where the color of the digits is correlated with their labels, introducing potential bias in the learning process. This setup encourages models to rely on color-based cues rather than shape-based ones. The task becomes particularly challenging when this correlation is disrupted, making it an ideal benchmark for testing generalization under distribution shifts.

**WaterBirds** is a binary classification task involving two types of birds (e.g., waterfowl and land birds), each associated with a distinct environmental background. The dataset evaluates a model's ability to handle spurious correlations in the training data, where the model may incorrectly rely on background cues (e.g., water vs. land) instead of the bird species itself. The primary challenge is for the model to generalize when the background context changes, forcing it to focus on more invariant, semantically meaningful features.

## E. Related Work

### E.1. Incremental Learning

Incremental learning (Van de Ven et al., 2022; Wu et al., 2019; Castro et al., 2018) algorithms are designed to handle non-stationary data, where the data distribution evolves over time. A widely used method in this domain is the Sequential Karhunen–Loeve (SKL) algorithm, introduced by Levy and Lindenbaum (Levy & Lindenbaum, 1998). This algorithm efficiently updates both the eigenbasis and the sample mean as new data arrives, providing a dynamic and computationally efficient solution for continuously evolving data streams. A more refined variant of SKL was later proposed in (Lim et al., 2004), enhancing the subspace update mechanism, particularly when the subspace mean plays a crucial role in the learning task.

One prominent application of incremental learning is in motion estimation, where systems need to adapt to temporal variations in visual data. This is particularly relevant in video processing and visual tracking, where observations change from frame to frame. In such scenarios, the subspace constancy assumption is often employed, suggesting that the low-dimensional subspace capturing the object's appearance or motion remains approximately stable over short time intervals. This assumption allows for incremental updates, avoiding the need for complete retraining, thus improving both the efficiency and stability of motion tracking across frames.

The incremental PCA algorithm (Ross et al., 2008) builds upon SKL by incorporating a forgetting factor. This factor gradually down-weights the influence of older observations, ensuring that more recent data has a stronger impact on the model than outdated samples. This mechanism is crucial for maintaining the model's relevance in dynamic environments, particularly when dealing with large-scale, continuously evolving data streams.

### E.2. Contrastive Learning

Contrastive learning (Khosla et al., 2020; Wang & Qi, 2022), an unsupervised approach, focuses on learning effective feature representations by constructing positive and negative sample pairs, enabling the model to distinguish between samples from different categories. Recent advancements in deep learning have driven significant progress in contrastive learning, particularly in areas such as computer vision, natural language processing, and speech processing.

MoCo (He et al., 2020) further enhanced the efficiency and stability of contrastive learning by maintaining a dynamic queue of negative samples. Using a momentum-based update mechanism, MoCo addresses the challenge of selecting negative samples during training, significantly improving the quality of learned representations. By increasing the distance between the original sample and its negative counterparts in the feature space, the model can learn more accurate representations.

Beyond its success in unsupervised learning, contrastive learning has also paved the way for new methodologies in multi-modal learning, transfer learning, and other related fields.

### E.3. Test-Time Adaptation

Test-Time Adaptation (TTA) (Lin et al., 2025; Yoo et al., 2025) is an online learning paradigm that adapts deep neural networks (DNNs) pretrained on a source domain to unseen target distributions at test time, aiming to improve robustness under distribution shifts. Recent TTA studies mainly focus on self-supervised adaptation, where only a subset of model parameters is updated without access to source data or its statistics (Lee et al., 2024; Niu et al., 2023; 2022). Tent (Wang et al., 2021) relates prediction entropy to distribution shift and minimizes entropy to update Batch Normalization (BN) layers for adaptation. Building on Tent, EATA (Niu et al., 2022) mitigates noisy gradients induced by high-entropy samples by actively selecting reliable and non-redundant test instances. DeYO (Lee et al., 2024) further argues that entropy alone is an unreliable confidence indicator, and proposes object-destructive transformations to filter predictions that are more grounded in object shape. Beyond these BN-based methods, Candidate Pseudolabel Learning (CPL) (Zhang et al., 2024) introduces a prompt-tuning framework for vision–language models using unlabeled data, where a candidate label set is constructed via intra- and inter-instance confidence-based selection and optimized under a partial-label learning formulation. To avoid discarding potentially useful test samples, PTTA (Ma et al., 2025a) proposes a plug-and-play strategy that purifies malicious samples rather than filtering them out by retrieving benign samples with opposite effects on the adaptation objective and applying Mixup-based purification, thereby improving test-time data utilization and online adaptation stability.

### E.4. Continual Test-Time Adaptation

Continual Test-Time Adaptation (CoTTA) (Wang et al., 2022; Gan et al., 2023; Han et al., 2025; Zhang et al., 2025b) extends standard TTA to a more realistic setting where the target distribution evolves over time in a non-stationary stream. To cope with this continual shift, CoTTA (Wang et al., 2022) employs an EMA teacher–student framework to provide more stable pseudo-labels and enforces consistency under data augmentations, thereby alleviating error accumulation during online adaptation. It further introduces a stochastic restoration mechanism that randomly resets a small portion of parameters toward the source model, which helps prevent long-term drift and mitigates catastrophic forgetting.

Motivated by the observation that real-world test streams can be correlated and contaminated by diverse noise while undergoing continual shifts, STAF (Xiong et al., 2024) pushes TTA towards practical noise scenarios by explicitly stabilizing the adaptation process. Specifically, STAF introduces multi-stream perturbation consistency to enforce weak-to-strong view consistency, a reliable memory-based corrector to calibrate predictions using reliable snapshots, and a dynamic parameter

restoration strategy that adapts the restoration probability based on the shift/adaptation degree, jointly mitigating error accumulation and catastrophic forgetting.

Beyond pseudo-label based adaptation, CMAE (Liu et al., 2024a) reduces the reliance on noisy supervision by adopting a masked autoencoder based self-supervised framework for continual adaptation. CMAE performs distribution-aware masking by selectively masking high-uncertainty tokens and enforcing prediction consistency between original and masked views, and additionally reconstructs robust features instead of raw pixels to improve invariance against corruptions and domain shifts. This combination enables more stable adaptation under non-stationary target streams while reducing error accumulation.

To explicitly balance short-term adaptation and long-term knowledge retention, ViDA (Liu et al., 2024b) proposes a homeostatic visual domain adapter that decomposes adaptation into a low-rank branch for preserving domain-shared knowledge and a high-rank branch for capturing domain-specific variations. Moreover, ViDA introduces a homeostatic knowledge allotment mechanism that dynamically adjusts the contribution of the two branches based on prediction uncertainty, effectively mitigating both error accumulation and catastrophic forgetting in continual adaptation.

To better balance instantaneous discrimination and long-term generalization under continual shifts, Navigating CTTA (Yang et al., 2024) proposes a dual-stream framework that learns symbiotic knowledge by tuning only normalization parameters in one stream while updating all learnable parameters in the other. It further injects source knowledge via weighted soft parameter alignment and improves supervision quality by calibrating pseudo-labels with source priors, selecting reliable labels using self-adaptive confidence thresholds, and employing a soft-weighted contrastive module to recover potential semantics, thereby reducing noisy updates and catastrophic forgetting.

