# OpenReview forum: "SaTeen: Learning Structural Alignment for Continual Test-Time Adaptation"
_ICML.cc/2026/Conference — ICML 2026 regular_

### Official Review · Reviewer_ywfe · 2026-02-21

**Soundness:** 3
**Presentation:** 2
**Significance:** 3
**Originality:** 2
**Overall Recommendation:** 4
**Confidence:** 5

**Summary:**

This paper investigates the vulnerability of entropy minimization (EM) to noisy data in continual test-time adaptation (CTTA). It highlights the importance of structural information to avoid model collapse. Accordingly, it proposes SaTeen, which introduces an Intra-Sample Structural Alignment objective that regularizes EM by maximizing the predictive cross-entropy between an input and its structure-disrupted counterpart. Furthermore, it employs Inter-Sample Structure Alignment that filters reliable samples to construct the intrinsic data manifold via Incremental Principal Component Analysis (IPCA), which regularizes the adaptation by minimizing the feature reconstruction error of all test samples. Extensive experiments demonstrate that SaTeen achieves higher performance compared to several baselines.

**Compliance With Llm Reviewing Policy:**

Affirmed.

**Final Justification:**

My concerns have been adequately addressed. I'll keep my positive score.

**Key Questions For Authors:**

Please refer to the Weaknesses.

**Limitations:**

yes

**Strengths And Weaknesses:**

Strengths
1. The paper provides a solid theoretical analysis, which makes it easy to understand the vulnerability of entropy minimization (EM) to noisy data in continual test-time adaptation (CTTA).
2. The paper comprehensively validate the proposed SaTeen across diverse and challenging scenarios (e.g., continual TTA, single TTA, and extreme imbalanced label shifts) on multiple benchmarks.
3. The proposed SaTeen is intuitively straightforward, and the source code in the supplementary material significantly enhances the reproducibility of this work.

Weaknesses
1. The definition of "structural features" is ambiguous.
2. The motivation of this paper bears a resemblance to the recently proposed Decoupled Entropy Minimization [1]. Both works critically identify the flaws of entropy minimization (EM) in test-time adaptation and introduce a structurally similar negative penalty term to the loss function to prevent model collapse. However, this manuscript omits theoretical discussion, or empirical comparison with this highly relevant work. This omission significantly weakens the claimed novelty.
3. In the Inter-Sample Structure Alignment (Sec.3.2), since the selected "reliable samples" are ready to construct the intrinsic manifold of the test data, it is intuitively unclear why the $\mathcal{L}_2$ reconstruction error constraint must be universally applied to all test data. A more logical approach might be to use the manifold constructed from reliable samples to specifically constrain and guide the "unreliable samples" that are prone to deviation. The theoretical necessity for imposing this uniform constraint across all global data is not adequately justified. Moreover, the "reliable samples" is selected by a threshold, and it could also include noisy data. Under this situation, why the constructed low dimensional PCA subspace still works well?
4. The Intra-Sample and Inter-Sample Structure Alignments are currently lack of connections. Why is the combination of both strictly necessary to achieve optimal performance? Do they offer mathematical complementarity in the feature space or during gradient updates?
5. During deployment, the model cannot perform grid search for hyperparameter tuning on the target domain beforehand. Given this, how should the critical hyperparameters ($\lambda_1$ and $\lambda_2$) be determined a priori during actual deployment?
6. Current TTA research is rapidly expanding to vision-language foundation models (e.g., CLIP, BLIP, SigLIP). Since the proposed intra-sample alignment heavily relies on "patch shuffling"—a purely vision-specific destructive operation—it remains unclear whether this mechanism is applicable to multi-modal architectures that heavily depend on aligned image-text semantic spaces. Discussing or providing preliminary experiments on the applicability of SaTeen to VLMs would significantly broaden the paper's contribution to the community.

[1] Ma et al., Decoupled Entropy Minimization, NeurIPS-2025.

---

> ### Author Rebuttal · Authors · 2026-03-31
>
> Thanks for the helpful comments. We address the main concerns below.
>
> **W1:** For the image classification tasks studied in this paper,  the term **structural features** is **explicit**. Specifically, **intra-sample structure** refers to the relatively stable spatial organization within a single image, such as the object’s overall shape, part relationships, and layout. **Inter-sample structure** refers to the shared low-dimensional geometry of a set of samples in feature space, i.e., the intrinsic manifold that should remain stable as the test stream evolves over time.
>
> **W2:** Although both SaTeen and DEM analyze model collapse through entropy decoupling, SaTeen provides a more fundamental explanation by studying the model’s **gradients and update directions**, thereby showing why the proposed loss can mitigate collapse. In contrast, DEM mainly suppresses model collapse by leveraging **statistical priors**. We additionally include experiments on adaDEM under several settings. The results are as follows:
>
> | Method     | TTA  | CTTA | TTA-IB | CTTA-IB |
> | ---------- | ---- | ---- | ------ | ------- |
> | VitBase-LN | 44.2 | 44.2 | 28.5   | 28.5    |
> | Tent       | 48.1 | 49.0 | 47.8   | 51.5    |
> | adaDEM     | 60.2 | 60.7 | 59.7   | 50.8    |
> | SaTeen     | 65.4 | 64.6 | 64.7   | 61.0    |
>
> From these results, we observe that when **the label prior does not change (TTA, CTTA, TTA-IB)**, the gap between adaDEM and SaTeen is relatively small. However, under **CTTA with imbalanced labels**, especially when **the prior itself is also drifting**, adaDEM degrades significantly.
>
> The reason is that **adaDEM** corrects model predictions by estimating a prior distribution from past predictions, but in **continual TTA** this prior is not necessarily stationary. Once **the prior itself also shifts**, this mechanism becomes unreliable, and the method may fail.
>
> **W3:** The manifold is constructed from past reliable samples, while the new incoming reliable samples are used to optimize $L_2$. We consider an equilibrium case: when the feature space no longer updates, i.e., $M_t = M_{t-1}$, then $Q_t$ is a zero matrix. In this case, $Q_t$ and $L_2$ are equivalent in essence, one is in matrix form, while the other is in sample form. Therefore, the subsequent reliable samples should also be aligned.
>
> **W4:** From the **definition of structure itself**, **intra-sample structure** refers to the relatively stable spatial organization within a single image, such as the object’s overall shape, part relationships, and layout. **Inter-sample structure** refers to the shared low-dimensional geometry of a set of samples in feature space, i.e., the intrinsic manifold that should remain stable as the test stream evolves over time.
>
> From the **objective** perspective, **intra-sample alignment** acts on the **local** update of each sample and helps avoid model collapse. **Inter-sample alignment** acts on the **global** geometric stability of the test stream and mitigates catastrophic forgetting.
>
> From the **mathematical form**, $L_1$ operates on the **prediction distribution**, changing the discriminative update direction for each sample; $L_2$ operates on the reconstruction error of the **feature subspace**, constraining geometric drift in the representation space.
>
> From the **experimental results**, intra-sample alignment is the most important single component. However, adding inter-sample alignment on top of it further improves performance, and this gain is more pronounced in **continual TTA**, indicating that the inter-sample module mainly complements long-term forgetting/stability issues in sequential adaptation.
>
> **W5:** Our goal is not target-specific hyperparameter optimization, but a robust default configuration. The same $(\lambda_1, \lambda_2) $ is used across datasets, and the sensitivity study shows that SaTeen remains stable over a broad range of values.
>
> **W6:** Following the setting of adaDEM, we conducted the following experiment:
>
> | Method          | ImageNet (IN) | IN-A | IN-V2 | IN-R | IN-Sketch | Average |
> | --------------- | ------------: | ---: | ----: | ---: | --------: | ------: |
> | CLIP-ResNet50   |          58.2 | 21.8 |  51.4 | 56.2 |      33.4 |    44.2 |
> | TPT+CoOp        |          65.4 | 28.9 |  58.2 | 59.0 |      36.3 |    49.6 |
> | adaDEM+CoOp |          65.6 | 31.3 |  58.5 | 59.3 |      36.3 |    50.2 |
> | SaTeen+CoOp   |          65.5 | 32.1 |  58.8 | 59.6 |      36.4 |    50.5 |
>
> Since prompt tuning does not update the image encoder, **inter-sample alignment is not used** in this setting. The results show that **SaTeen also performs well on VLMs**.
>
> Finally, we would like to thank the reviewers again for their valuable comments on our paper!

---

> > ### Author Rebuttal · Reviewer_ywfe · 2026-04-01
> >
> > Thanks for the authors' effort on the response. I'll keep my score.

---

> > > ### Author Response · Authors · 2026-04-01
> > >
> > > Dear Reviewer ywfe,
> > >
> > > Thank you again for your thoughtful evaluation and for acknowledging the novelty and contributions of our work. We also greatly appreciate your constructive feedback.
> > >
> > > Should you have any further comments or suggestions, we would be very grateful and will respond as carefully as possible.
> > >
> > > Best regards,
> > >
> > > The Authors

---

### Official Review · Reviewer_vMA4 · 2026-02-24

**Soundness:** 3
**Presentation:** 3
**Significance:** 3
**Originality:** 3
**Overall Recommendation:** 4
**Confidence:** 4

**Summary:**

This paper proposes a test-time adaptation (TTA) method, Structural Alignment-based Test-Time Adaptation (SaTeen), to address the model collapse of existing entropy-based TTA methods.
When a distribution shift is severe, a model tends to perform imbalanced classification because it focuses on noise features rather than structural ones.
SaTeen uses patch-shuffled images to contrast noisy features with structural features, enabling the model to adapt to them during TTA.
In addition, SaTeen constructs a PCA basis from reliable test sample features and fits the features within the subspace to stabilize TTA.
Experimental results show that SaTeen improved classification accuracy in biased and image-corruption settings, including continual TTA.

**Compliance With Llm Reviewing Policy:**

Affirmed.

**Final Justification:**

The authors addressed my concerns and showed that a general negative sample selection strategy works.
I would like to update my score to 4.

**Key Questions For Authors:**

- Q1: How were the hyperparameters selected?
- Q2: Can SaTeen be applied to non-image data?
- Q3: SENTRY is an unsupervised domain adaptation method rather than TTA. How was it rendered to the TTA setting?

**Limitations:**

Limitations should be discussed.

**Strengths And Weaknesses:**

## Strengths
- **S1**: Theoretical analysis on model collapse in the imbalanced prediction regime is intriguing.
- **S2**: The paper is well-written.
- **S3**: Addressing the model collapse, which is common in the standard entropy-based TTA approaches, is highly relevant.

## Weaknesses
- **W1**: The title and introduction are too general; SaTeen is only applicable to image data due to the patch shuffling. Explicitly limiting the domain to image classification would be beneficial.
- **W2**: Adding more discussion on the difference between SaTeen and DeYO would strengthen the uniqueness of SaTeen since DeYO also uses patch shuffling for distinguishing structural features from noise features.
- **W3**: Empirical validation of the theoretical results would be more persuasive. Theorem 3.1 states that SaTeen can avoid the bad parameter update direction that leads to model collapse. It would be interesting if this is observed also in the experiment. For example, comparing the gradients of the naive entropy, the proposed loss, and a supervised loss (e.g., cross entropy) could verify the theorem.
- **W4**: While SaTeen attempts to minimize the intra-sample loss, the proposed final loss in Eq. (18) weights the samples by $\alpha$ based on the intra-sample loss itself, i.e., samples with high intra-sample loss are not learned. Is the form of the final loss in Eq. (18) effective? While the cross entropy ranges in $[0,+\infty)$ mathematically, confirming the actual loss distribution and how the weighting and filtering work would be beneficial.
- **W5**: Unifying the baselines among experiments would be more convincing. For instance,
    - ReCAP is included in Tab. 2, but not in Tab. 1.
    - SENTRY is included in Tab. 1, but not in Tab. 2.
- **W6**: Visualizing features with PCA would reflect the feature structure more directly since the inter-sample structure alignment is based on PCA.
- **W7** (minor): Some grammatical errors should be corrected. For instance,
    - Left column, L99: `empirically results`
    - Right column, L149: `we divides`
- **W8** (minor): Tab. 4 is not mentioned in any main paragraph.

---

> ### Author Rebuttal · Authors · 2026-03-31
>
> Thanks for the helpful comments. We address the main concerns below.
>
> **Q1:** $\lambda_1$ balances EM and CE. Since EM is the main driver of adaptation, while CE mainly strengthens feature focus and prevents model collapse, we set it to a value in $[0,1]$, namely 0.4.
>
> $\lambda_2$ balances an entropy-form loss and the $L_2$-norm loss; this weight is typically small, so we set it to 0.05.
>
> $k$ is the PCA subspace dimension. We set it to about 5% of the feature dimension.
>
> $\tau$ follows prior methods.
>
> **Q2, W1:** In principle, data of any modality have structure and can benefit from structural alignment. For example, **inter-sample alignment** is modality-agnostic. Although **intra-sample alignment** requires negative samples, such negatives can be naturally defined for many common modalities.
>
> **Q3:** We evaluate SENTRY in its original UDA setting, training the model with unlabeled test data for adaptation, and evaluate it on the test set. We include it as baselines to show that SaTeen still performs better under a stricter source-free setting.
>
> **W2:** DeYO mainly focuses on **sample selection**, whereas SaTeen’s **intra-sample alignment** is not just a filter, but also an **explicit training objective**. It both encourages the model to focus on each sample’s **structural features** and suppresses **model collapse**.
>
> **W3:** This is a great suggestion. To validate Theorem 3.1, we compare the cosine similarity between the gradients of the proposed loss and supervised CE, and between the gradients of the entropy and supervised CE. The results are as follows:
>
> | Method  | Gauss  | Shot   | Impl   | Defoc  | Glass  | Motion | Zoom   | Snow   | Frost  | Fog    | Brit   | Contr  | Elastic | Pixel  | JPEG   |
> | ------- | ------ | ------ | ------ | ------ | ------ | ------ | ------ | ------ | ------ | ------ | ------ | ------ | ------- | ------ | ------ |
> | Tent    | -0.009 | -0.019 | -0.027 | -0.030 | -0.035 | -0.035 | -0.036 | -0.041 | -0.046 | -0.054 | -0.052 | -0.051 | -0.057  | -0.054 | -0.053 |
> | Entropy | 0.001  | 0.004  | 0.006  | -0.003 | -0.004 | -0.002 | -0.001 | -0.002 | -0.004 | -0.010 | -0.005 | -0.001 | -0.002  | 0.008  | 0.001  |
> | SaTeen  | 0.004  | 0.007  | 0.011  | 0.011  | 0.014  | 0.018  | 0.018  | 0.020  | 0.020  | 0.014  | 0.019  | 0.023  | 0.027   | 0.034  | 0.037  |
>
> Here, **Tent** denotes the cosine similarity between the gradient of naive entropy and that of supervised cross-entropy under Tent adaptation. **Entropy** and **SaTeen** denote the cosine similarities between supervised cross-entropy and the gradients of naive entropy and SaTeen loss, respectively, under SaTeen adaptation.
>
> The results show that under Tent, the entropy gradient is often opposite to the supervised gradient. Under SaTeen, the entropy gradient has low cosine similarity with the supervised one (gradient cancellation between reliable and unreliable samples). In contrast, the gradient induced by the SaTeen loss is generally aligned with the supervised gradient.
>
> **W4:** High-loss samples are unreliable, and applying EM to them is often harmful (EM is a form of unsupervised learning). On reliable samples, the CE term mainly prevents model collapse; on unreliable ones, its benefit is only marginal. Thus, minimizing this loss on high-loss samples is unnecessary. We further analyze the CE term distribution and the corresponding accuracy as follows:
>
> | CE loss       | 0.0-4.0 | 4.0-5.0 | 5.0-5.5 | 5.5-6.0 | 6.0-6.5 | 6.5-$\infty$ |
> | ------------- | ------- | ------- | ------- | ------- | ------- | ----------- |
> | Accuracy (%)  | 0.13    | 0.17    | 0.23    | 0.24    | 0.33    | 0.38        |
> | Frequency (%) | 25.3    | 21.7    | 15.7    | 28.4    | 7.3     | 1.6         |
>
> We observe that:
> (1) Higher CE usually corresponds to higher source-model accuracy; that is, when the model relies more on structural cues, predictions tend to be more accurate. A similar trend holds for entropy: lower-entropy samples are usually more accurate.
> (2) A small fraction of samples have extremely high CE, but their accuracy gain is limited.
>
> **First**, we minimize the SaTeen loss only on lower-loss samples, which are more reliable, and assign larger weights to more reliable samples. **Second**, since a few samples have extremely high CE without much extra accuracy gain, we cap the weight to avoid over-emphasizing them.
>
> **W5:** Due to missing dataset-specific files, we cannot reproduce ReCAP on Waterbirds or ColoredMNIST. SENTRY is infeasible on ImageNet mainly because the dataset is too large and retraining from scratch would be prohibitively expensive.
>
> **W6:** PCA also shows a similar trend and can be used for visualization. We use t-SNE because it better reveals the manifold structure. The PCA result is in:https://anonymous.4open.science/r/SaTeen_PCA_vi-0A5C/SaTeen_vi_PCA.pdf
>
> **W7,W8:** We will address these minor weaknesses.
>
> Finally, we would like to thank the reviewers again for their valuable comments on our paper!

---

> > ### Author Rebuttal · Reviewer_vMA4 · 2026-04-01
> >
> > I appreciate the authors' response and additional experiments.
> >
> > The comparison of gradient similarities is quite interesting.
> >
> > > Q2, W1: In principle, data of any modality have structure and can benefit from structural alignment. For example, inter-sample alignment is modality-agnostic. Although intra-sample alignment requires negative samples, such negatives can be naturally defined for many common modalities.
> >
> > In patch shuffling of an image, SaTeen assumes that noise features are separated from structural ones, not just for making negative samples.
> > Are there specific data processing procedures for other modalities?
> >
> >
> > -----
> >
> > After reply rebuttal comment
> >
> > I appreciate the authors' further response and additional experiments.
> > My concerns have been resolved.
> > I updated my score.

---

> > > ### Author Response · Authors · 2026-04-03
> > >
> > > Dear Reviewer vMA4,
> > >
> > > We apologize for not responding sooner, as we were busy conducting additional experiments.
> > >
> > > We fully understand your concern and would like to clarify the following. Noisy features can be divided into **sample-wise** and **domain-wise** noise. Our main focus is on **domain-wise noisy features**, because in the TTA setting each sample is used for adaptation only once, so errors caused by sample-wise noisy features do not accumulate. In contrast, over-reliance on **domain-wise** noisy features can accumulate over time and eventually lead to **model collapse**.
> > >
> > > Separating inter-sample structural features from **all** noisy features is relatively difficult. However, separating them only from **domain-wise noisy features** is much easier. For example, one can select a sample from another class in the current domain (this sample no longer preserves the original structural features, but still retains the domain-wise noisy features).
> > >
> > > **We aim to answer this question experimentally: SaTeen only requires negative samples to disrupt structural features, and such negatives do not need to rely on specific data processing procedures; they can be obtained in simpler and more general ways.**
> > >
> > > The experimental results are as follows:
> > >
> > > |                              | TTA        | CTTA       | TTA-IB      | CTTA-IB    |
> > > | ---------------------------- | ---------- | ---------- | ----------- | ---------- |
> > > | patch shuffing augmentation       | 49.9 / 1.4 | 47.1 / 1.3 | 50.7 / 2.7  | 50.4 / 3.1 |
> > > | random sample (in the batch) | 49.8 / 1.5 | 48.8 / 1.2 | 33.0 / 27.0 | 40.2 / 2.9 |
> > > | memory bank (out the batch)  | 47.8 / 1.7 | 46.7 / 1.9 | 48.3 / 3.3  | 49.1 / 3.3 |
> > >
> > > Here, **random sample (in the batch)** means randomly selecting another sample from the current batch as the negative sample, while **memory bank (out of the batch)** means maintaining a bank of samples from the current domain as negative candidates.
> > >
> > > We already reported the **random sample** results in the main paper. It achieves performance similar to **patch shuffling** in both **TTA** and **CTTA**. However, under **imbalanced labels**, a batch often contains samples from the same class, so this negative-sample strategy can fail.
> > >
> > > We therefore further introduce a **more general negative-sample selection strategy** that is not limited to the current batch. Specifically, we maintain an online-updated **memory bank** and select the negative sample.
> > >
> > > Next, we **describe the design of this memory bank in detail**:
> > >
> > > **Initialization:** To keep the design simple and avoid stage-specific operations, we fully initialize the bank as soon as the first batch arrives. Samples are added one by one: at each step, we insert the sample with the **lowest average cosine similarity** to the current bank, until the bank reaches size $N_{mb}$. Since the model is continuously adapting, we store the **raw images** rather than the features.
> > >
> > > **Negative-sample selection:** We randomly sample one example from the bank as the negative sample. This is the simplest strategy, and more sophisticated designs could further improve performance.
> > >
> > > **Insertion:** We select $N_{up}$ samples from the unreliable set ($N_{un}$) according to their average cosine similarity to the current bank, and insert them into the bank. We use **unreliable samples** because they were not used in the previous round of intra-sample alignment; if reliable samples were used instead, training would already have reduced their dependence on noisy features. Cosine similarity is used to avoid redundancy and maintain bank diversity.
> > >
> > > **Removal:** We compute the average cosine similarity of each current bank sample among the combined set of $N_{mb}+N_{up}$ samples, rank them, and then remove $N_{up}$ samples in total: $N_{up}-1$ are randomly removed from the **more similar half**, and 1 is randomly removed from the **less similar half**. This design avoids a failure mode in continual TTA where old samples can remain in the bank forever because they have very low similarity to the new domain. It also avoids introducing an additional **age-based** hyperparameter, which would be harder to tune.
> > >
> > > Since this design was implemented under limited time, we apologize if it is still imperfect. Nevertheless, we have already verified its effectiveness on ImageNet-C under **TTA**, **CTTA**, **TTA with imbalanced labels**, and **CTTA with imbalanced labels**.
> > >
> > > Thank you again for your thoughtful evaluation and for acknowledging the contributions of our work.
> > >
> > > We hope this rebuttal clarifies your concerns and addresses your questions. Should you have any further comments or suggestions, we would be very grateful and will respond as carefully as possible.
> > >
> > > Thank you for raising the score!
> > >
> > > Best regards,
> > > The Authors

---

### Official Review · Reviewer_Spin · 2026-02-26

**Soundness:** 3
**Presentation:** 3
**Significance:** 2
**Originality:** 2
**Overall Recommendation:** 4
**Confidence:** 4

**Summary:**

The paper proposes SaTeen, a test-time adaptation (TTA) and continual test-time adaptation (CTTA) framework designed to mitigate entropy-minimization–induced model collapse and catastrophic forgetting through the enforcement of two forms of structural alignment. The method demonstrates promising performance across multiple experimental settings on classification benchmarks.

**Compliance With Llm Reviewing Policy:**

Affirmed.

**Final Justification:**

Thank you to the authors for the detailed and thoughtful responses. Most of my concerns have been adequately addressed, and I have therefore decided to raise my score.

**Key Questions For Authors:**

1 How sensitive is SaTeen to the choice of patch shuffling as the structure-disrupting augmentation?

Have you tried other structural corruptions (e.g., cutout, blockwise randomization) and measured both performance and the behavior of δ / collapse? A short empirical comparison could clarify whether patch shuffling is essential or just convenient.

2 How many seeds and what variance did you use for Tables 3–4, 7–13?

Clarifying whether these are single-run results or averaged, and possibly adding stds, would increase confidence in the robustness of the reported SOTA.

**Limitations:**

Please refer to the above Strengths and Weaknesses section for comments regarding the limitations of the work.

**Strengths And Weaknesses:**

Strengths:

(1) The proposed intra-sample loss is simple yet well-motivated, and can be readily integrated into existing TTA pipelines with minimal modification.

(2) The inter-sample IPCA-based alignment is computationally efficient and well-suited for streaming TTA scenarios.

(3) Extensive quantitative experiments, together with thorough theoretical analysis, provide strong empirical and conceptual support for the effectiveness of the proposed method.

Weaknesses:

(1) Although the proposed intra-sample structural alignment is empirically effective, prediction consistency across augmented views has become a common strategy in the TTA literature. The manuscript does not sufficiently clarify what fundamentally distinguishes this formulation from existing augmentation-based regularization or consistency constraints, nor what constitutes its core novelty.

(2) The generalizability of the proposed framework remains unclear. The experiments are limited to classification benchmarks, and it would be important to evaluate whether the method can be extended to more complex dense prediction tasks, such as semantic segmentation or object detection. Demonstrating effectiveness in such settings would substantially strengthen the claim of methodological universality.

(3) The related work section lacks discussion and comparison with several closely related CTTA approaches whose motivations and mechanisms overlap with the current method. In particular, the omission of the following works is notable and should be addressed:

SATA: Source Anchoring and Target Alignment Network for Continual Test-Time Adaptation

PAID: Pairwise Angular-Invariant Decomposition for Continual Test-Time Adaptation

Continual Test-Time Domain Adaptation

EcoTTA: Memory-Efficient Continual Test-Time Adaptation via Self-Distilled Regularization

A clearer positioning against these methods is essential for assessing the paper’s relative contribution.

(4) The overall novelty appears incremental in light of prior structural and sample-selection–based CTTA methods. The intra-sample loss can be viewed as a combination of entropy minimization and cross-entropy consistency under augmentation, conceptually related to contrastive-style regularization and approaches such as DeYO that leverage object-destructive transformations for sample filtering. Similarly, the inter-sample IPCA alignment shares conceptual similarities with covariance alignment in LinearTCA and structural preservation mechanisms in PAID. While the integration of these components is interesting, the manuscript does not convincingly articulate how the proposed “structural alignment for CTTA” is conceptually distinct from, or substantially advances beyond, existing structural alignment paradigms. This point is particularly critical given that structural alignment constitutes the central novelty claim of the paper.

---

> ### Author Rebuttal · Authors · 2026-03-31
>
> Thanks for the helpful comments. We address the main concerns below.
>
> **Q1:** Here we provide more complete results beyond the ablation study. Here, **random** means randomly sampling another image from the batch, and **fft** means removing the low-frequency signal. The results are as follows:
>
> | Acc (%) / IR (%) |        TTA |       CTTA |      TTA-IB |     CTTA-IB |
> | ---------------- | ---------: | ---------: | ----------: | ----------: |
> | random           | 49.8 / 1.5 | 48.8 / 1.2 | 33.0 / 27.0 |  40.2 / 2.9 |
> | cutout           | 44.0 / 2.7 | 44.8 / 1.5 |  45.1 / 2.8 | 34.2 / 25.4 |
> | fft              | 42.3 / 2.5 | 43.3 / 2.3 |  44.6 / 3.0 | 35.1 / 22.6 |
> | patch            | 49.9 / 1.4 | 47.1 / 1.3 |  50.7 / 2.7 |  50.4 / 3.1 |
>
> Overall, **patch shuffling works best for object classification**.
>
> **Q2:** We used three seeds. In Tables 3-4, 7-13, the variances are all small, and we will add std in the revision.
>
> **W1:** Prior augmentation-consistency methods treat multiple augmentations as positive views and enforce prediction consistency. Instead, we (1) use a destructive view as a negative sample to prevent learning noisy features, where the negative can come from destructive augmentation or other choices such as randomly sampling another image from the current batch; and (2) **carefully design** the discrepancy loss to suppress model collapse. Theoretically, we have:
>
> $\frac{\partial H(\mathbf{x})}{\partial \theta}=-\sum(\log p_i(\mathbf{x})+ 1)\frac{\partial p_i(\mathbf{x})}{\partial \theta},$
>
> $-\frac{\partial \mathrm{CE}\left(sg[\mathbf{p}(\tilde{\mathbf{x}})]  \middle\| \mathbf{p(\mathbf{x})} \right)}{\partial \theta} = -\sum (- \frac{{p}_i(\tilde{\mathbf{x}})}{p_i(\mathbf{x})}\frac{\partial p_i(\mathbf{x})}{\partial \theta} )$
>
> $-\frac{\partial \mathrm{CE}\left(\mathbf{p(\mathbf{x})}  \middle\| sg[\mathbf{p(\tilde{\mathbf{x}})]} \right)}{\partial \theta}=-\sum ( - \log p_i(\tilde{\mathbf{x}}) \frac{\partial p_i(\mathbf{x})}{\partial \theta} )$
>
> where $-\frac{p_i(\tilde{\mathbf{x}})}{p_i(\mathbf{x})} < 0$ and $-\log p_i(\tilde{\mathbf{x}}) > 0$. By Theorem 3.1 (1 , 2), under collapse the gradient is dominated by the direction that increases $p_c$, so we focus on $\frac{\partial p_c(\mathbf{x})}{\partial \theta}$. Since training uses reliable (high-confidence) samples, $\left(\log p_c(\mathbf{x}) + 1\right) > 0$. Therefore, the first CE term can suppress collapse, while the second cannot.
>
> We also compare different methods experimentally.
>
> | Method      | Fog      | Avg(15 corruptions) |
> | ----------- | -------- | ------------------- |
> | DeYO        | 3.5  | 42.4                |
> | cos sim     | 3.1  | 39.3                |
> | $L_2$       | 2.7  | 43.3                |
> | $CE(p\|p')$ | 2.3  | 43.3                |
> | $CE(p'\|p)$ | **60.3** | 49.9                |
>
> Only $\mathrm{CE}(p' \| p)$ suppresses model collapse. Novelty is discussed in **W4**.
>
> **W2:** Due to time limits, we have not yet implemented SaTeen for other tasks, but we would be happy to present the results in the next stage.
>
> **W3:** CoTTA is addressed in **W1**. **EcoTTA** and **SATA** reduce forgetting and improve long-term stability via multi-level self-distillation and source-model anchoring, respectively. However, both are essentially **conservative methods**, which inherently limit adaptation. In contrast, SaTeen’s inter-sample alignment anchors evolving target representations to the **data structure**, rather than the **source model**. **PAID** is discussed in **W4**.
>
> **W4: Intra-sample novelty.**  To our knowledge, we are the **first to introduce negative samples for learning in CTTA**. Our key insight is that under **severe shift**, augmentation often **destroys structural information instead of enriching it**. This fundamentally differs from prior augmentation-consistency methods that enforce agreement across views, and may also inspire other shift settings. Moreover, the discrepancy measure is **carefully designed**: we are the first to show that $\mathrm{CE}(p' \| p)$ can suppress **EM-induced model collapse** from a gradient perspective. This insight may extend to other **multi-view EM** settings.
>
> **Inter-sample novelty.** The idea of invariance itself is classical; the challenge is **how to realize it under strict CTTA constraints**. Existing methods are not directly applicable: **LinearTCA** needs the **full test set**, while **PAID** uses **source-domain statistics**. Our method makes structural alignment naturally compatible with **source-free, online CTTA**. This gives an alignment mechanism that is both **efficient** and **effective**.
>
> **Framework novelty.** More importantly, we are the **first method to jointly model both intra- and inter-sample structure**. Prior methods usually consider only **one side**, and thus address only part of the CTTA problem. In our method, the two are complementary.
>
> Finally, we would like to thank the reviewers again for their valuable comments on our paper!

---

> > ### Author Rebuttal · Reviewer_Spin · 2026-04-03
> >
> > Thank you to the authors for the detailed and thoughtful responses. Most of my concerns have been adequately addressed, and I have therefore decided to raise my score.

---

> > > ### Author Response · Authors · 2026-04-03
> > >
> > > Dear Reviewer Spin,
> > >
> > > We fully understand your concern and would like to clarify the following. We do **not** assume that **reliable-sample selection is perfect**. We analyzed the prediction accuracy of samples under different intra-sample loss ranges:
> > >
> > > | Intra-loss    | -5~1 | 1~2 | 2~3 | 3~7 |
> > > | ------------- | ----: | ---: | ---: | ---: |
> > > | Accuracy (%)  |  51.1 | 20.4 |  8.8 |  2.8 |
> > > | Frequency (%) |  20.6 | 17.5 | 35.6 | 26.3 |
> > >
> > > As shown above, even the group with the highest initial accuracy still has only moderate accuracy (51.1). In other words, in actual training, many unreliable samples still participate in intra-sample structural alignment, yet SaTeen still performs well.
> > >
> > > From a theoretical perspective, for **intra-sample alignment**, only samples identified as reliable are used for this training term, because entropy-style updates on high-loss samples are often unstable or even harmful. For **inter-sample alignment**, reliable samples are not treated as perfectly correct supervision; instead, they are used collectively to estimate a shared subspace/manifold. In this case, a small amount of noisy samples is averaged out in the overall geometric estimation, and its effect is usually much smaller than in per-sample direct supervision. That is, the inter-sample module is naturally more robust to limited selection noise.
> > >
> > > Second, **our method does not rely absolutely on $\tau$**. As shown by **(Tent,3,12,15)** in the ablation studies (Tables 5 and 16 in the main paper). Comparing **Tent** with **(3)** under CTTA shows that **inter-sample alignment** can still help in the continual setting even **without** $\tau$. Comparing **Tent**, **(12)**, and **(15)** further shows that, even without using $\tau$ for filtering, SaTeen still works and remains much stronger than the Tent baseline.
> > >
> > >
> > > | ResNet50-TTA | $\lambda_1$ | $\lambda_2$ | $\tau$ | $\alpha$ | Avg. |
> > > | ----------------------------- | -: | -: | -: | -: | ---: |
> > > | Tent                          |    |    |    |    | 30.6 |
> > > | (2)                           |    |    | ✓  |    | 32.5 |
> > > | (3)                           |    | ✓  |    |    | 30.9 |
> > > | (4)                           | ✓  |    |    |    | 31.0 |
> > > | (5)                           |    | ✓  | ✓  |    | 32.0 |
> > > | (6)                           | ✓  |    | ✓  |    | 34.4 |
> > > | (7)                           | ✓  | ✓  |    |    | 39.5 |
> > > | (8)                           | ✓  | ✓  | ✓  |    | 39.7 |
> > > | (9)                           |    |    |    | ✓  | 30.7 |
> > > | (10)                          |    |    | ✓  | ✓  | 34.0 |
> > > | (11)                          |    | ✓  |    | ✓  | 27.7 |
> > > | (12)                          | ✓  |    |    | ✓  | 42.5 |
> > > | (13)                          |    | ✓  | ✓  | ✓  | 34.5 |
> > > | (14)                          | ✓  |    | ✓  | ✓  | 47.1 |
> > > | (15)                          | ✓  | ✓  |    | ✓  | 42.2 |
> > > | SaTeen                        | ✓  | ✓  | ✓  | ✓  | **49.9** |
> > >
> > > | ResNet50-CTTA. | $\lambda_1$ | $\lambda_2$ | $\tau$ | $\alpha$ | Avg. |
> > > | -------------------------------- | -: | -: | -: | -: | ---: |
> > > | Tent                             |    |    |    |    |  8.2 |
> > > | (2)                              |    |    | ✓  |    | 34.2 |
> > > | (3)                              |    | ✓  |    |    | 23.4 |
> > > | (4)                              | ✓  |    |    |    | 27.3 |
> > > | (5)                              |    | ✓  | ✓  |    | 36.8 |
> > > | (6)                              | ✓  |    | ✓  |    | 38.5 |
> > > | (7)                              | ✓  | ✓  |    |    | 31.5 |
> > > | (8)                              | ✓  | ✓  | ✓  |    | 42.3 |
> > > | (9)                              |    |    |    | ✓  |  8.2 |
> > > | (10)                             |    |    | ✓  | ✓  | 13.5 |
> > > | (11)                             |    | ✓  |    | ✓  | 32.4 |
> > > | (12)                             | ✓  |    |    | ✓  | 40.3 |
> > > | (13)                             |    | ✓  | ✓  | ✓  | 37.0 |
> > > | (14)                             | ✓  |    | ✓  | ✓  | 43.2 |
> > > | (15)                             | ✓  | ✓  |    | ✓  | 34.8 |
> > > | SaTeen                           | ✓  | ✓  | ✓  | ✓  | **47.1** |
> > >
> > > Thank you again for your thoughtful evaluation and for recognizing the contributions of our work.
> > >
> > > We hope this rebuttal clarifies your concerns and addresses your questions. Should you have any further comments or suggestions, we would be very grateful and will respond as carefully as possible.
> > >
> > > Thank you for raising the score!
> > >
> > > Best regards,
> > > The Authors

---

### Official Review · Reviewer_143p · 2026-03-12

**Soundness:** 3
**Presentation:** 3
**Significance:** 3
**Originality:** 2
**Overall Recommendation:** 4
**Confidence:** 3

**Summary:**

This paper primarily tackles the model collapse problem under class-imbalanced settings in test-time adaptation and additionally mitigates catastrophic forgetting in continual adaptation settings. The authors propose a Structural Alignment-based Test-Time Adaptation method (SaTeen), which consists of an intra-sample loss that contrasts the predictions of a clean input with its structure-disrupted version and an inter-sample loss that uses incremental PCA to preserve structural consistency across evolving domains. Experimental results on ImageNet-C, ColoredMNIST, and WaterBirds across various test-time adaptation settings demonstrate that SaTeen consistently outperforms other methods.

**Compliance With Llm Reviewing Policy:**

Affirmed.

**Final Justification:**

I have increased my score to a Weak Accept based on the authors' responses. I trust that the authors will fulfill their commitment to incorporating the additional experiments and discussions—specifically regarding the ablation study of patch shuffling strategy on more diverse datasets—into the final version of the manuscript.

**Key Questions For Authors:**

- Would a class-balance regularization objective (e.g., maximizing the entropy of the average prediction) be effective in mitigating class collapse? Furthermore, would such a regularization term be complementary to SaTeen’s current intra-sample design?

- The inter-sample alignment assumes that the underlying structural manifold of the feature space remains stable across different domains. Does this assumption hold in scenarios involving large domain gaps? Under what specific conditions would this structural alignment fail to preserve semantic consistency?

**Limitations:**

A dedicated limitation and failure analysis is currently missing. Including such a discussion would significantly clarify the application boundaries of the method.

**Strengths And Weaknesses:**

**Strengths**

- The focus on the model collapse problem in entropy-based test-time adaptation (TTA) is significant and practically relevant, especially as real-world deployments often encounter class-imbalanced data.

- The methodology design is complementary; the intra-sample loss effectively addresses class collapse, while the inter-sample loss mitigates catastrophic forgetting, positioning SaTeen as a versatile framework for various TTA settings.

- The experimental evaluation is extensive, covering diverse scenarios such as single TTA, continual TTA, mixed-shifts, and imbalanced labels across a wide range of benchmarks (ImageNet-C, ImageNet-R, ColoredMNIST, WaterBirds, and VisDA).


**Weaknesses**

- SaTeen adopts patch shuffling as the default strategy for generating negative samples; however, the paper does not explore other potentially effective alternatives, such as frequency-domain augmentation, style perturbation, or masking strategies.

- Since patch shuffling inherently disrupts spatial object structure, its effectiveness is primarily demonstrated on object-centric recognition tasks. Its performance on non-object-centric tasks—such as scene classification (where global layout matters) or texture-based classification—remains unstudied and unverified.

- The framework introduces four manually tuned hyperparameters ($\lambda_1$, $\lambda_2$, $k$, and $\tau$) without providing a clear selection protocol or guidance. While these values are fixed across the reported datasets, it remains unclear how the specific values (e.g., 0.4 and 0.2) were determined and whether they will remain effective when the model is applied to entirely new datasets or domains.

---

> ### Author Rebuttal · Authors · 2026-03-30
>
> Thanks for the helpful comments. We address the main concerns below.
>
> **Q1:** This is a very interesting question. We conducted the following experiment: we minimized the entropy of each individual sample prediction, while maximizing the entropy of the average prediction over the current batch. The results are as follows:
>
> | Acc (%) / $\delta$ (%)      |          TTA |          CTTA |       TTA-IB |       CTTA-IB |
> | --------------------- | -----------: | ------------: | -----------: | ------------: |
> | ResNet50-GN           |   30.6 / 6.4 |    30.6 / 6.4 |   31.0 / 6.4 |    31.0 / 6.4 |
> | Tent                  |  33.5 / 31.2 |    8.2 / 85.2 |  22.8 / 39.2 |    5.1 / 89.4 |
> | Tent + maxE           |   37.8 / 3.5 |    37.6 / 2.5 |   20.1 / 5.4 |     7.7 / 5.4 |
> | $\triangle$                | +4.3 / -25.7 | +29.4 / -82.7 | -2.7 / -33.8 |  +2.6 / -84.0 |
> | SaTeen (Intra)        |   47.1 / 1.9 |    43.2 / 1.8 |   48.2 / 3.5 |    46.9 / 3.2 |
> | SaTeen (Intra) + maxE |   48.3 / 1.2 |    44.5 / 1.3 |   23.3 / 4.7 |   10.4 / 13.9 |
> | $\triangle$                 |  +1.2 / -0.7 |   +1.3 / -0.5 | -24.9 / +1.2 | -36.5 / +10.7 |
>
> We draw two conclusions:
>
> **(1)** Class-balance regularization can reduce imbalanced predictions.
>
> **(2)** When the **label prior is uniform (TTA,CTTA)**, maximizing the entropy of the average prediction can improve model performance and can be complementary to SaTeen’s current intra-sample design. However, when the **label prior is non-uniform and shifts over time(TTA-IB,CTTA-IB)**, this term can severely hurt performance.
>
> **Q2:** When the domain gap is sufficiently large, the **assumption may fail**. A representative example is the spurious correlation setting in the Waterbirds dataset. We conducted the following experiment:
>
> | Waterbirds               | Avg Acc | Worst-Group Acc |
> | ------------------------ | ------: | --------------: |
> | ResNet18-BN              |    84.3 |            67.0 |
> | Tent                     |    83.2 |            54.5 |
> | SaTeen (Inter-only + AS) |    81.2 |            48.3 |
> | SaTeen (Inter-only + AR) |    86.5 |            68.4 |
>
> Here, **AS means** aligning to the source samples, while **AR means** aligning to the reliable samples. In this experiment, the intra-sample loss is used only for sample selection, not for training.
>
> Direct AS performs worse, especially on the Worst Group, because source and target may rely on very different features; this gap is even larger for worst groups. If the source model itself learns spurious cues, AS can be harmful. In contrast, AR remains beneficial: the reliable-sample subspace is **updated online**, so later samples are closer to it than to the source subspace, and reliable samples usually depend on more accurate features.
>
> Thus, even under large domain shift, **inter-sample alignment** remains effective if the shift is gradual. It may fail when the when the **domain changes rapidly, or when the reliable-sample selection mechanism fails**.
>
> **W1:** Here we provide more complete results beyond the ablation study. Here, **random** means randomly sampling another image from the batch, and **fft** means removing the low-frequency signal. The results are as follows:
>
> | Acc (%) / $\delta$ (%) |        TTA |       CTTA |      TTA-IB |     CTTA-IB |
> | ---------------- | ---------: | ---------: | ----------: | ----------: |
> | random           | 49.8 / 1.5 | 48.8 / 1.2 | 33.0 / 27.0 |  40.2 / 2.9 |
> | mask             | 44.0 / 2.7 | 44.8 / 1.5 |  45.1 / 2.8 | 34.2 / 25.4 |
> | fft              | 42.3 / 2.5 | 43.3 / 2.3 |  44.6 / 3.0 | 35.1 / 22.6 |
> | patch            | 49.9 / 1.4 | 47.1 / 1.3 |  50.7 / 2.7 |  50.4 / 3.1 |
>
> Overall, **patch shuffling works best for object classification**.
>
> **W2:** The notion of **structure is task-dependent**. For object classification, the key structure is shape and spatial layout, so patch shuffling is appropriate. For scene or texture classification, patch shuffling may be ineffective, but other perturbations can be used to destroy the task-specific structural features. We would be happy to include such results in the discussion.
>
> **W3:**
> $\lambda_1$ balances EM and CE. Since EM is the main driver of adaptation, while CE mainly strengthens feature focus and prevents model collapse, we set it to a value in $[0,1]$, namely 0.4.
>
> $\lambda_2$ balances an entropy-form loss and the $L_2$-norm loss; this weight is typically small, so we set it to 0.05.
>
> $k$ is the PCA subspace dimension. We set it to about 5% of the feature dimension.
>
> $\tau$ follows prior methods.
>
> We cannot guarantee effectiveness on all datasets. Our goal is not target-specific tuning, but **a robust default setting**. The same $(k,\lambda_1,\lambda_2,\tau)$  is used across datasets, and the sensitivity study shows stable performance over a broad range.
>
> Finally, we would like to thank the reviewers again for their valuable comments on our paper!

---

> > ### Author Rebuttal · Reviewer_143p · 2026-04-04
> >
> > I have increased my score to a Weak Accept based on the authors' responses. I trust that the authors will fulfill their commitment to incorporating the additional experiments and discussions—specifically regarding the ablation study of patch shuffling strategy on more diverse datasets—into the final version of the manuscript.

---

> > > ### Author Response · Authors · 2026-04-04
> > >
> > > Dear Reviewer 143p,
> > >
> > > Thank you for raising the score!
> > >
> > >
> > >
> > > Due to time constraints, we may not be able to complete the relevant experiments and discussions within the next four days. However, we assure you that these will be included in the appendix of the final version of the manuscript.
> > >
> > >
> > >
> > > Should you have any further comments or suggestions, we would be very grateful and will respond as carefully as possible.
> > >
> > >
> > >
> > > Best regards,
> > >
> > >
> > >
> > > The Authors

---

### Decision · Program_Chairs · 2026-04-30

**Decision:**

Accept (regular)

**Comment:**

This paper proposes Structural Alignment-based Test-Time Adaptation (SaTeen), a continual TTA method designed to address the model collapse issues of existing entropy-based approaches. The authors did a good job during the rebuttal, and all reviewers ultimately recommended acceptance. Thus, the AC can easily recommend acceptance. However, the authors are strongly encouraged to include the additional experiments and discussions in the final version.